# Alk1 acts in non-endothelial VE-cadherin⁺ perineurial cells to maintain nerve branching during hair homeostasis

Gopal Chovatiya [1], Kefei Nina Li[1], Jonathan Li[1], Sangeeta Ghuwalewala[1] & Tudorita Tumbar [1] ✉

Vascular endothelial (VE)-cadherin is a well-recognized endothelial cell marker. One of its interacting partners, the TGF-β receptor Alk1, is essential in endothelial cells for adult skin vasculature remodeling during hair homeostasis. Using single-cell transcriptomics, lineage tracing and gene targeting in mice, we characterize the cellular and molecular dynamics of skin VE-cadherin⁺ cells during hair homeostasis. We describe dynamic changes of VE-cadherin⁺ endothelial cells specific to blood and lymphatic vessels and uncover an atypical VE-cadherin⁺ cell population. The latter is not a predicted adult endovascular progenitor, but rather a non-endothelial mesenchymal perineurial cell type, which forms nerve encapsulating tubular structures that undergo remodeling during hair homeostasis. Alk1 acts in the VE-cadherin⁺ perineurial cells to maintain proper homeostatic nerve branching by enforcing basement membrane and extracellular matrix molecular signatures. Our work implicates the VE-cadherin/Alk1 duo, classically known as endothelial-vascular specific, in perineurial-nerve homeostasis. This has broad implications in vascular and nerve disease.

Vascular endothelial (VE) cadherin (encoded by *Cdh5*) has been traditionally known as an endothelial cell (EC) specific marker, expressed by both blood vessels (capillaries, veins, and arteries) and lymphatic vessels[1]. The encoding gene and mRNA are referred to as *Cdh5* and the protein product as VE-cadherin, a convention we follow throughout our paper. VE-cadherin is a cell-cell contact molecule, recognized for its essential role in the formation and maturation of vascular tubes[1,2]. It also contributes to signaling within the ECs by interacting with receptors of the VEGF, Notch, FGF, and the TGF-β pathway, including Alk1, a TGF-β receptor expressed by ECs[3]. Its molecular interactions ensure vasculature resilience, flexibility, mobility, and permeability, thus endowing ECs with their specific characteristics[3–6]. Nonendothelial tumor cells can also aberrantly activate VE-cadherin, forming tubular irrigation networks that assist tumor growth[7]. Single-cell (sc) RNA-seq analysis of ECs from multiple adult organs (but not skin) uncovered significant tissue-specific gene expressions[8–10], but generally identified the same type of mature ECs. Intriguingly, single-cell

transcriptomics in the adult mouse dorsal aorta uniquely identified a distinct *Cdh5*⁺ cellular subset proposed to be an immature endothelial vascular progenitor (EVP)[11]. A VE-cadherin⁺ EVP was also suggested to arise in skin wounds and tumors[12,13], but it has not yet been reported in normal skin. To date, single-cell transcriptomics have not yet characterized non-EC types within the normal VE-cadherin expressing lineage. In addition, there is little knowledge of the dynamic changes in VE-cadherin lineage heterogeneity and molecular makeup during adult tissue homeostasis, in the absence of injury or cancer.

Adult skin and hair follicles are an excellent model system to study the organization of adult VE-cadherin⁺ lineages during tissue homeostasis. Skin is a highly regenerative tissue that requires an ongoing supply of O₂ and nutrients. The hair follicles, made largely of epithelial cells, undergo periodic transitions from resting/quiescence (telogen) to growth (anagen) phase during the hair homeostatic cycle. These phases are synchronous in mice for the first cycle, providing an opportunity to access homeostatic processes that are generally more

[1]Department of Molecular Biology and Genetics, Cornell University, Ithaca, NY, USA. ✉e-mail: tt252@cornell.edu

hidden in less synchronous adult tissues. At anagen, the skin hypodermis significantly expands in size and becomes filled with engorging adipocytes and blood vessels that wrap around the downward growing hair follicle bulbs, while the lymphatic vessels also remodel their caliber and change their drainage capacity[14,15] (Fig. 1a). Alk1, the known VE-cadherin interacting partner, acts in all skin ECs to regulate their migration and the vasculature remodeling during hair cycle[16]. Other skin structures, such as nerves and sub-cutaneous muscles, may also be remodeled during the hair cycle[17]. A thorough characterization of the cellular and functional heterogeneity of skin VE-cadherin[+] lineage organization during adult skin homeostasis is currently lacking.

Here, we employed 10x Genomics scRNA-seq of sorted *Cdh5*-CreERT[2];tdTomato marked cells isolated from adult mouse skin at two hair cycle stages: quiescence (telogen) and growth (anagen). This data combined with genetic lineage tracing and Alk1 gene targeting in specific skin cell types allows us to refine and redefine the cellular and functional heterogeneity of the VE-cadherin[+] lineage during adult tissue homeostasis. We rule out a potential VE-cadherin[+] endovascular progenitor (EVP) that may be responsible for vasculature growth during hair cycle. Moreover, we firmly expand the VE-cadherin lineage attributes and the function of Alk1 from vascular-endothelial into other cell types and functions, including perineurial-nerve remodeling. This work opens new avenues of investigation into vascular and nerve remodeling functions in adult regenerative tissues, with potential implications to nerve and vascular diseases.

## Results

### Labeling and isolation of adult skin VE-cadherin[+] (*Cdh5*) cells documents turnover rates

Skin vasculature remodels during the hair cycle[18–20], as adult skin considerably expands in thickness from telogen to anagen[21] (Fig. 1a). At anagen, the hair follicle grows downward into the hypodermis, which fills up with fat cells, while the blood vessels (BV) vasculature stretches out and nourishes tissue growth[16,22]. A horizontal vascular plexus (CD31[+]/VE-cadherin[+]) underneath the hair germ (HPuHG) made mostly of BV disperses out into the hypodermis at the telogen-anagen transition[16,18] (Fig. 1a). Whole mount skin imaging[16] and 70 μm skin sections that were VE-cadherin stained or *Cdh5*-CreERT[2] x tdTomato (*Cdh5*-tdTomato[+]) labeled, followed by 3D confocal microscopy confirmed these findings (Fig. 1b and Supplementary Fig. 1a, b). Aside from BVs, the lymphatic capillaries also remodel by increasing in caliber and moving slightly away from the HFs[23,24].

We and others also reported an increase in Ki67[+] or BrdU[+] proliferative CD31[+] EC at anagen by immunofluorescence of skin sections[18–20]. To quantitatively evaluate the magnitude of VE-cadherin[+] skin lineage proliferation in homeostasis, we performed BrdU labeling from postnatal (PD)17-25, encompassing the hair cycle transition from quiescence to growth. We used *Cdh5*-CreERT[2] x tdTomato (*Cdh5*-tdTomato[+])[25,26] reporter mice induced with tamoxifen (TM) at PD17, followed by FACS isolation of tdTomato[+] ECs at PD25 (Supplementary Fig. 1c, d). Confocal microscopy of skin sections confirmed both high BrdU labeling efficiency and the presence of BrdU[+]/VE-cadherin[+] vascular cells in single optical images (Supplementary Fig. 1e). Analysis of FACS sorted tdTomato[+] cells deposited on slides and stained with antibodies for BrdU and VE-cadherin (*Cdh5*) (Supplementary Fig. 1f) showed that ~80% of the sorted tdTomato[+] were strongly VE-cadherin-positive by immunofluorescence (IF) staining, and of those ~15% incorporated BrdU during the 8-day labeling period (Supplementary Fig. 1g). This suggests that in the period that accompanies the transition from hair quiescence to growth ~ 1-2% ECs divide on average/day. Proliferation was reported to occur through early anagen and to peak at full anagen[18–20]. Thus, we can estimate that at least one-third of skin VE-cadherin-expressing cells engage in proliferation during the first hair cycle in mice. This data quantitively documents the extent of

cellular turnover in the VE-cadherin[+] skin compartment during the hair cycle (Supplementary Fig. 1h).

### scRNA-seq profiling of *Cdh5*-CreERT[2] marked skin cells uncovered VE-cadherin (*Cdh5*)[+] endothelial and non-endothelial lineages

To understand the cellular heterogeneity and the dynamic molecular reorganization of VE-cadherin[+] cells in the hair cycle, we performed 10x Genomics scRNA-seq analysis of *Cdh5*-CreERT[2] x tdTomato[+]; *Krt14*-H2BGFP[-] cells purified from mouse dorsal skin at quiescence (telogen) and growth (anagen) stages (Fig. 1c–e and Supplementary Fig. 1a), using methods previously described[27]. Two mouse replicates (S1 and S2) from each stage were subjected to 10x scRNA-seq library construction and sequencing (see Methods, GEO accession number GSE211381). After applying quality control parameters on individual samples (Supplementary Fig. 1i), we obtained a total of 7250 and 9262 high-quality cells at telogen and anagen, respectively. All samples were integrated using the *Harmony* package (see Methods), followed by Uniform Manifold Approximation and Projection (UMAP) analysis which produced 16 distinct cell populations from the tdTomato[+]/GFP[-] cells, highly reproducible in all 4 mice analyzed (Fig. 1f and Supplementary Fig. 1j). Based on a panel of known lineage markers (Figs. 1g), 7 populations were ECs and showed robust *Cdh5* expression (Fig. 1h). Previously characterized EC type-specific markers revealed distinct populations of artery, vein, capillaries, and lymphatics[28–30] (Fig. 1h). The capillaries (Cap) displayed two (*Cxcl12*[+]/*CD36*[+]) populations enriched in either Vegfr2 (*Kdr*) or *Rgcc*[9,28]. The lymphatics were either collecting vessels (Lymph-Col, *Prox1*[+]) or capillaries (Lymph-Cap, *Ccl21a*, *Pdpn* and *Lyve1*[+]). The Ki67[+] proliferative (Proli) EC population was ~1-2% of the total, which agrees with our BrdU data (Fig. 1f, g and Supplementary Fig. 1g).

Apart from these 7 endothelial populations, 6 populations were *Cdh5* negative contaminating immune cells and fibroblasts, while 2 populations that expressed low levels of *Cdh5* were vascular smooth muscle cells (vSMCs) or pericytes and schwann cells (Fig. 1h and Supplementary Data 1). Surprisingly, one population expressed medium *Cdh5* levels and *Krt19*, previously reported as an exclusive HFSC marker[31]. This *Cdh5*[+]/*Krt19*[+] population represented over 15% of all sorted cells (Fig. 1f), did not share expression of mature ECs and could not be classified as pericyte, schwann, or fibroblast cells. Together, these data provide high-quality transcriptomic profiles for adult VE-cadherin[+] skin lineages at different hair cycle stages and identified known mature endothelial populations as well as a few not previously reported populations that were likely not endothelial.

### VE-cadherin[+] endothelial populations undergo dynamic redistribution and gene expression changes during the hair cycle

We first focused on the vascular EC clusters to probe in more depth the dynamics of cellular and molecular heterogeneity during the hair cycle. The 7 ECs populations from the VE-cadherin marked cells, represented 2543 (telogen) and 7498 (anagen) high-quality sequenced cells (Supplementary Fig. 2a) with high correlation between replicates (Supplementary Fig. 2b). Previous single-cell analysis suggested a tissue-specific marker expression of ECs[9], thus we examined individual population enriched genes and identified skin-specific marker combination of ECs (Supplementary Fig. 2c, d and Supplementary Data 2).

UMAP analysis showed that all 7 EC populations that include the proliferative ECs are present at both telogen and anagen (Fig. 2a), with no emergence or loss of any EC type. Interestingly, the relative fraction of each population changed at anagen when compared to telogen, with a near doubling of blood EC populations at the expense of the lymphatic populations (Fig. 2b). To test for this change in the intact tissue, we co-stained skin sections at telogen and anagen with CD31 and either Endomucin (blood vessel - BV- marker) or Prox1 (lymphatic vessel - LV- marker). Using 8 μm thin sections to reduce cell overlap, we

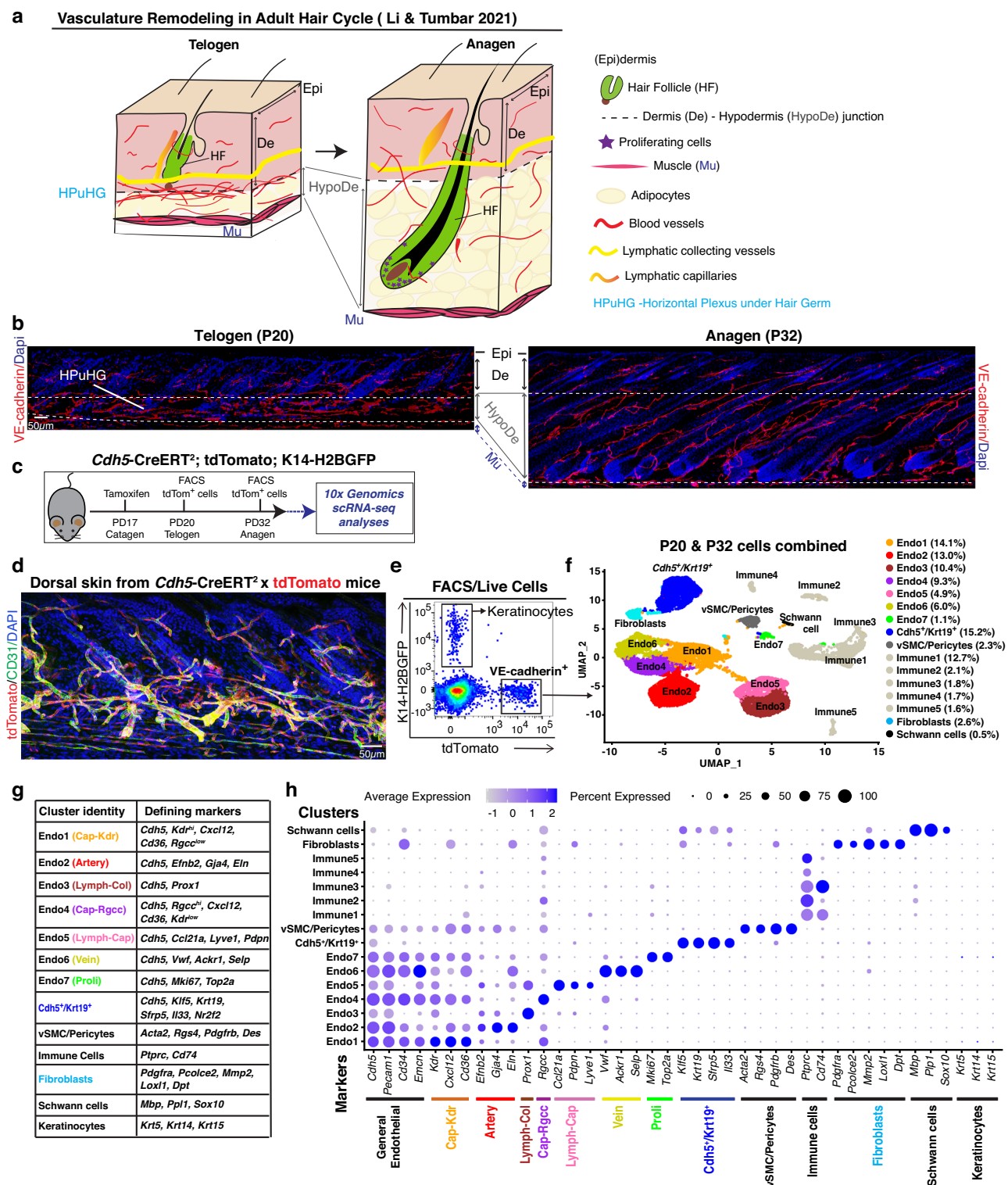

**Fig. 1 | Endothelial and non-endothelial adult skin VE-cadherin-marked cells characterized by scRNA-seq. a** Schematic highlighting skin and vasculature remodeling during adult mouse hair cycle[15]. **b** Maximal projections of confocal optical z-stacks of 70 μm thick dorsal skin sections stained with VE-cadherin (red) and Dapi (blue) at telogen and anagen. See also Supplementary Fig. 1a, b. Scale bar 50 μm. Multiple adjacent images were combined using the stitching function in Fiji. (n = 3 biologically independent samples). **c** Schematic showing tamoxifen induction and sample collection timepoints of *Cdh5*-CreERT²; tdTomato; *Krt14*-H2BGFP mice for scRNA-seq analysis. **d** Immunofluorescence imaging of *Cdh5*-tdTomato (red) in dorsal skin tissue sections stained for CD31 (green), scale bar 50 μm. (n = 3 biologically independent samples). **e** FACS purification of tdTomato⁺ cells from dorsal skin of *Cdh5*-CreERT² x tdTomato; *Krt14*-H2BGFP mice, n = 2 mice/stage as the VE-cadherin⁺ lineage. **f** Uniform Manifold Approximation and Projection (UMAP) plot generated after combining all 4 tdTomato⁺ sorted samples (2 samples/stage) showing all 16 cell populations, their first level identity and their relative abundance in %. All four samples were integrated using the *Harmony* package before UMAP analysis. See also Supplementary Fig. 1i, j. **g** Table of previously known markers that were used for population identification. **h** Dotplot showing expression of the population identifying markers.

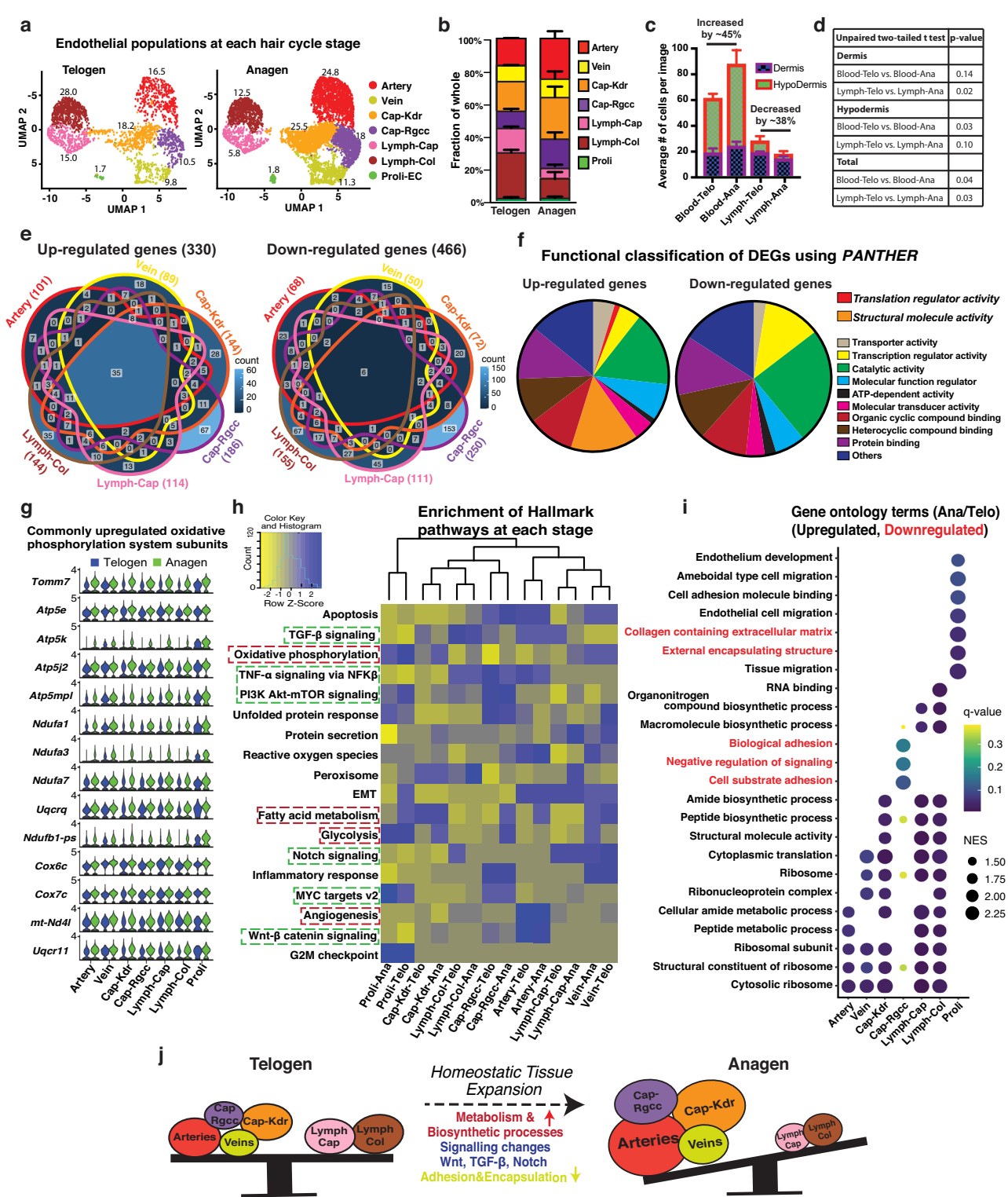

identified nuclei enclosed by CD31 signal and counted BV and LV EC density in both dermis and hypodermis (Supplementary Fig. 2e, f). The data showed a significant increase in BV EC density by ~45% at anagen, mainly in the hypodermis area (Fig. 2c, d). On the contrary, the LV EC counts showed decreased density at anagen mostly detected in the dermis (Fig. 2c, d). This in vivo validation by IF counts confirmed the population fractional change between the hair cycle rest and growth phases predicted by the scRNA-seq data.

The changes in BV/LV ECs density suggested either active conversions between these EC types or input of new ECs from an active

progenitor. To gain more insight into the dynamics of the EC populations, we examined the gene expression changes during the two critical stages. Many genes changed expression (log2FC > 0.25) in each EC population at the telogen-anagen transition (Supplementary Fig. 3a). Differentially expressed genes (DEGs) (330 upregulated and 466 downregulated or 784 total unique genes >1.3x) extracted from the nonproliferative EC populations (Fig. 2e and Supplementary Data 3) revealed diverse functional categories via *PANTHER*[32] analysis (Fig. 2f), while heatmap clustering revealed DEG changes from telogen to anagen (Supplementary Fig. 3b). Interestingly, the Cap-Kdr

**Fig. 2 | Cellular and molecular dynamics in all endothelial populations at the telogen-to-anagen transition. a** UMAP plot showing 7 EC populations detected at telogen and anagen stages. **b** Fraction of each EC population (from total 100%) in scRNA-seq datasets showing increase of blood vessel proportion at anagen, n = 2 mice/stage. Data are presented as mean ± SD. Source data are provided as a Source Data file. **c** Quantification of blood and lymphatic EC numbers in thin (8 μm) skin sections by IF staining of skin tissue with defined markers [CD31 (all EC) and either Endomucin (blood vessels) or Prox1 (lymphatic vessels)]. Unpaired student t-test was used to directly compare each compartment between the two stages (N ~ 30 images from n = 3 mice/stage). Data are presented as mean ± SD. Source data are provided as a Source Data file. **d** Table showing *p* value significance for graph in (**c**). An unpaired two-tailed t test was used for statistics. **e** Venn diagram for differentially expressed genes (DEGs) that changed expression at telogen-anagen transition

(FC > 1.3) in nonproliferating endothelial populations. Source data are provided as a Source Data file. **f** Functional gene categories of up-and down-regulated at anagen vs telogen using the *PANTHER* classification system. Source data are provided as a Source Data file. **g** Differential expression of oxidative phosphorylation system components at telogen and anagen. **h** Hallmark pathway analysis using all detected genes (raw counts obtained using *FetchData* function in *Seurat*) at each stage. The enrichment score of all hallmark pathways in all populations was compared by heatmap2 function in R. Source data are provided as a Source Data file. **i** Dotplot compilation of Gene Ontology (GO) terms analysis results performed using DEGs (FC > 1.3) at anagen. Red fonts indicate downregulated categories. **j** Cartoon summarizing cellular and functional dynamic changes in endothelial populations while they transit from telogen to anagen stage in vivo.

population clustered with the arteries and with the proliferative cells while the Cap-Rgcc population clustered with the veins (Supplementary Fig. 3b), suggesting potential sub-lineage relationships. Only a small fraction of 35 genes was upregulated in all EC populations at anagen, and those genes were involved in metabolism and oxidative phosphorylation, indicating increased cellular bioenergetics (Fig. 2g and Supplementary Data 3). On the other hand, *Hallmark* pathways[33] analysis of all detected genes show EC-population specific enrichment and changes between the stages (Fig. 2h, Supplementary Fig. 3c and Supplementary Data 4). Angiogenesis genes[34] were strongly enriched in arteries relative to the other clusters, along with related pathways, including Wnt, TGF-β, and TNF-α pathways (Fig. 2h and Supplementary Fig. 4), and these genes did not change at anagen. This observation, along with the BrdU incorporation at telogen/early anagen (Supplementary Fig. 1g), and the comparable percent of Ki67+ ECs at telogen and anagen (Fig. 2b) suggest that angiogenesis likely occurs in ECs at both stages, not just at anagen. Pseudotime analysis by Monocle[35] predicted the artery population as the most immature on the lineage trajectory, but a specific flow from one EC population into another was not clearly apparent (Supplementary Fig. 3d). EC populations other than arteries also showed changes in several signaling pathways previously implicated in vascular remodeling, such as TGF-β, NFK-β, PI3K-Akt-mTOR, Notch and MYC pathways (Fig. 2h). These were accompanied by changes in various gene ontology (GO) terms, which indicated overall vasculature growth and/or remodeling at anagen in all EC populations (Fig. 2i).

These data documented the EC heterogeneity in hair cycle and attested to the dynamic and intricate nature of cellular and molecular transformation that appear to implicate all EC populations during homeostasis (Fig. 2j). Despite significant shifts in the balance of EC types between stages, the analysis did not reveal a clear predicted flow of ECs from one population into another, nor did it uncover a common and highly active or changing immature vascular progenitor, as identified in skin injury and cancer[12]. This prompted us to further characterize the remaining VE-cadherin+ lineages that did not clearly qualify as ECs, in search for a potential EC vascular progenitor.

## VE-cadherin+ lineage contains a *Krt19*+ mesenchymal cell population outside the hair follicle

The remaining VE-cadherin (*Cdh5*) expressing populations that were not known ECs comprised of pericytes, schwann cells, and a previously unknown *Cdh5*+/*Krt19*+ population (Fig. 3a–c). VE-cadherin (*Cdh5*) levels were low when compared with the mature ECs (Fig. 3c), which may explain why previous analyses not based on the highly sensitive *Cdh5*-CreERT[2] x tdTomato labelling have missed this expression. Pericytes specifically expressed *Des*, *Acta2*, *Rgs4* and *Notch3*, as expected, and displayed enrichment of metabolism, biosynthetic pathways, and growth-related features at anagen, similar with the EC populations (Supplementary Fig. 5a, b). The schwann cell population displayed the known markers expression pattern (Supplementary Fig. 5c), and unlike all other VE-cadherin populations, it remained

unchanged in gene expression between the two hair cycle stages (Supplementary Data 5, adjusted *p*-val ns). The levels of VE-cadherin in this population were especially low and marked at most ~25% of cells in this population (Fig. 3e).

Finally, the *Cdh5*+/*Krt19*+ cluster expressed medium VE-cadherin (*Cdh5*) levels and captured a relatively large population that represents ~15% of all sorted cells (Fig. 3a). This population did not express CD31 (*Pecam1*) nor other mature EC markers (Fig. 1h), was highly distinct from all other *Cdh5*+ populations in UMAP analysis (Fig. 3b) and had its own defining set of marker expression (Fig. 3d, e). To probe the localization of this population in skin (and failing Keratin19 immunofluorescence staining), we employed the *Krt19*-CreERT; tdTomato mice induced with tamoxifen at PD17 and sacrificed at PD20 and PD49. As expected from previous studies[36], HFs showed prominent tdTomato+ labeling, but we also identified tdTomato+ signal in the dermis (Supplementary Fig. 5d). These non-HF structures stained clearly for VE-cadherin, though expression was weaker when compared with mature vasculature (Fig. 3f). GO analysis revealed a strong mesenchymal signature and enrichment in extracellular matrix (ECM) organization as well as structural assembly of an external encapsulating structure (Fig. 3g and Supplementary Fig. 5e). These features were common with some of the known *Cdh5*+ EC populations (Fig. 3h). In addition, many upregulated genes in the *Cdh5*+/*Krt19*+ population were uniquely associated with a mesenchymal identity that distinguishes these cells from known ECs (Supplementary Data 1) and are involved in adhesive functions such as collagen, heparin, and glycosaminoglycan binding (Fig. 3h). Together, these results confirmed the existence of a VE-cadherin (*Cdh5*)+/*Krt19*+ non-HF population in vivo, and revealed its strong and unique mesenchymal-type gene signature highly enriched in ECM secretion and binding.

## *Cdh5*+/*Krt19*+ population forms tubular structures but does not contribute to mature vasculature

Having defined the molecular characteristics of the *Cdh5*+/*Krt19*+ population, we then interrogated its physiological function during skin homeostasis. We first wondered if these cells may be the reported immature endovascular progenitors (EVPs), with low VE-cadherin expression[11–13]. Using a stringent set of genes specifically upregulated (FC > 4) in our *Cdh5*+/*Krt19*+ population relative to mature ECs, we found significant overlap with the reported aorta EVP signature (Fig. 4a) and markers (Fig. 4b and Supplementary Fig. 6a). Typical mature EC markers previously reported[11,12], such as CD31 and CD34 were downregulated in the *Cdh5*+/*Krt19*+ population (Fig. 4c and Supplementary Fig. 6b). Other observations consistent with a putative identity as EVPs of the *Cdh5*+/*Krt19*+ population were: an apparent fractional decrease from telogen to anagen in favor of the mature EC populations (Fig. 4d); differential gene expression changes in growth-associated genes (Fig. 4e); and suggestive Monocle lineage trajectory predictions placing the *Cdh5*+/*Krt19*+ population as the ground state (Fig. 4f and Supplementary Fig. 6c-d'). This prediction did not hold for a control fibroblast population (Supplementary Fig. 6e-f').

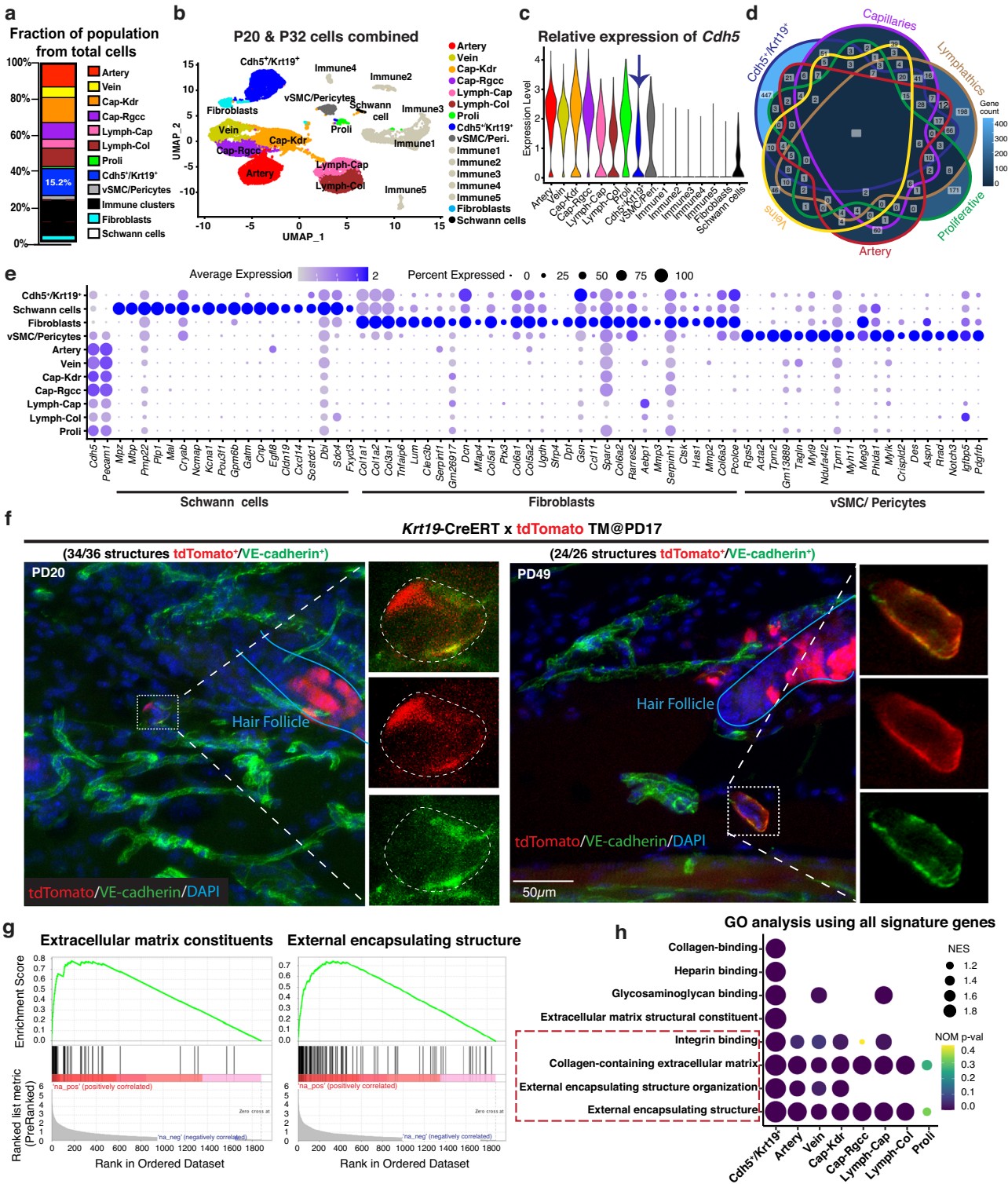

**Fig. 3 | The *Cdh5⁺/Krt19⁺* population identifies a non-hair mesenchymal VE-cadherin⁺ population. a** Cellular fraction of FACS isolated cells highlight comparative large size of *Cdh5⁺/Krt19⁺* population in scRNA-seq datasets. Source data are provided as a Source Data file. **b** UMAP plot with endothelial identity for comparative visualization with other panels. **c** Violin plot showing expression of VE-cadherin (*Cdh5*) in the isolated populations. Note reduced expression of *Cdh5* in *Cdh5⁺/Krt19⁺* population as compared to EC populations. **d** Venn diagram highlighting the unique molecular identity of the *Cdh5⁺/Krt19⁺* population as compared to blood and lymphatic endothelial populations with 447 DEGs. Source data are provided as a Source Data file. **e** Dotplot for highly expressed top genes in schwann cells, fibroblasts and vSMC/pericytes populations. **f** Co-immunofluorescence

imaging of *Krt19*-CreERT;tdTomato (red) and VE-cadherin antibody staining (green) on dorsal skin tissue sections at PD20 and PD49 showing co-localization of *Krt19*-tdTomato⁺ cells with VE-cadherin expressing cells, and their presence outside hair follicle in vivo. (*n* = 3 biologically independent samples). **g** Gene set enrichment analysis for *Cdh5⁺/Krt19⁺* population showing strong representation of extracellular matrix and encapsulation functions. Source data are provided as a Source Data file. **h** Comparison of top *Cdh5⁺/Krt19⁺* population GO categories revealed functional similarities with classical EC populations. Statistics on GO enrichment is based on the Kolmogorov–Smirnov test (KS test) for NES and the two-sided permutation test for p-values, as performed by the GSEA package[33]. Source data are provided as a Source Data file.

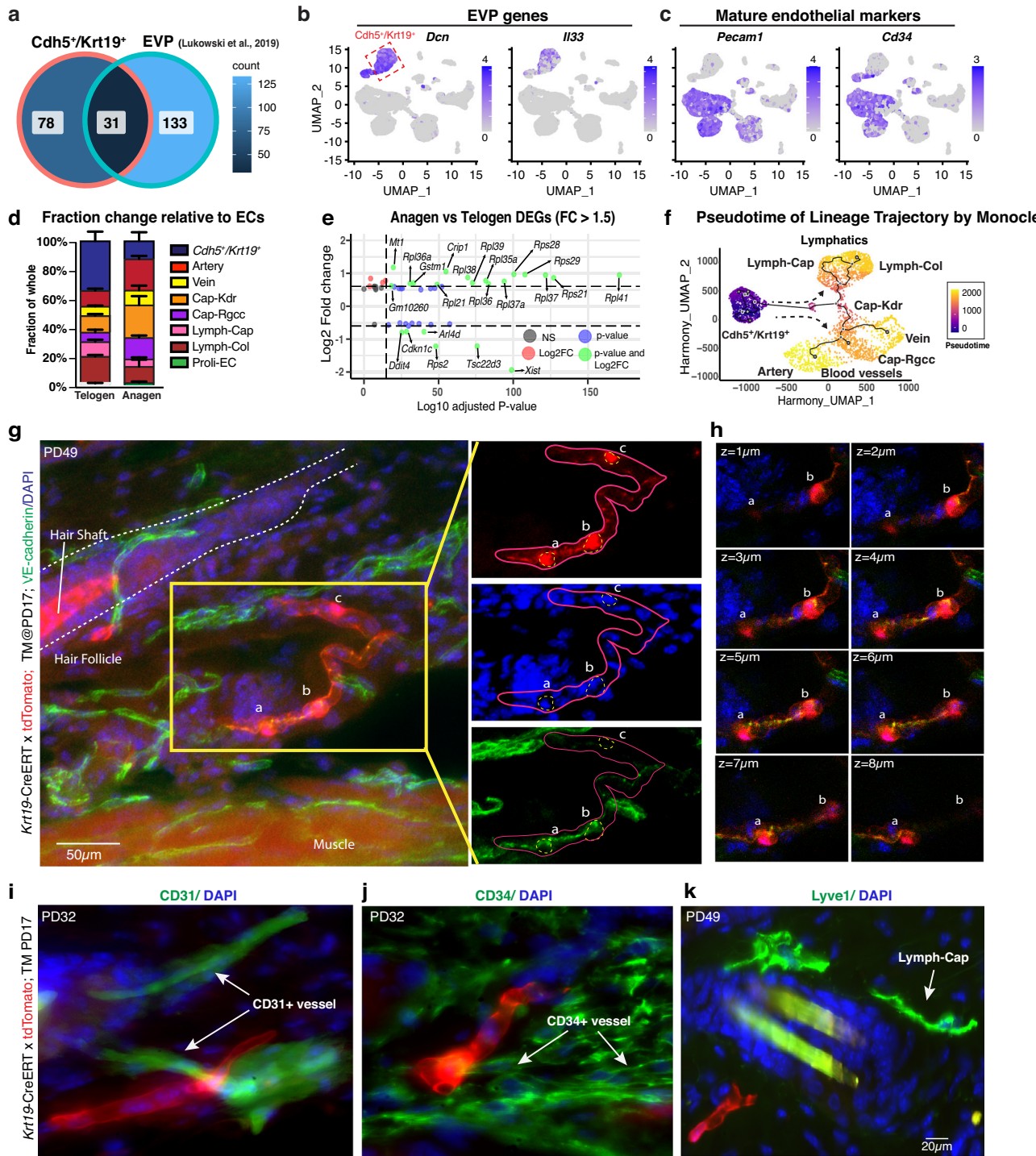

**Fig. 4 | *Cdh5⁺/Krt19⁺* cells do not contribute into the mature vasculature in vivo.**
**a** Venn diagram highlighting the overlap gene signatures between the previously characterized endovascular progenitor (EVP) population by Lukowski *et al.*, 2019[11] and our *Cdh5⁺/Krt19⁺* population (FC > 4). Source data are provided as a Source Data file. **b**, **c** Feature plots showing expression of indicated markers. **d** Fraction change of *Cdh5⁺/Krt19⁺* population along with endothelial populations in scRNA-seq datasets during telogen to anagen transition, *n* = 2 mice/stage. Data are presented as mean ± SD. Source data are provided as a Source Data file. **e** DEGs of the *Cdh5⁺/Krt19⁺* population between anagen and telogen are associated with protein translation and biosynthesis, suggesting increased activity. Statistics is based on a nonparametric Wilcoxon rank sum test performed by the Seurat package[73]. **f** Monocle3 predicted lineage trajectory and pseudotime ordering, which placed

the *Cdh5⁺/Krt19⁺* population as the putative common progenitor ground state.
**g**, **h** Confocal Z-stacks maximal projection (**g**) showing co-localization of *Krt19*-CreERT; tdTomato positive cells with VE-cadherin-stained cells at different optical planes (**h**) and their tubular structure outside hair follicle in dermis. Scale bar 50 µm. (*n* = 3 biologically independent samples). **i–k** Immunofluorescence staining of mature blood and lymphatic EC markers CD31 (**i**), CD34 (**j**) and Lyve1 (**k**) on skin sections from *Krt19*-CreERT;tdTomato mice. Tamoxifen was injected at PD17 and analyses were performed at indicated time points (*n* = 3 mice/stage) with *N* > 45 images per stain/stage. No colocalization was observed in all images analyzed indicating lack of contribution of the tdTomato⁺ structures into mature skin vasculature.

If the *Cdh5*⁺/*Krt19*⁺ cells were indeed skin EC vascular progenitors acting in the hair cycle, we reasoned they should eventually form mature vasculature. To test this, we employed lineage tracing using the *Krt19*-CreERT; tdTomato mice TM-induced at PD17 and stained the tissue sections at PD32 (anagen) and PD49 (telogen) with VE-cadherin, CD31 (*Pecam1*), CD34 and lymphatic capillary marker Lyve1 in *n* = 3 mice per stage. Excitingly, we did observe tdTomato⁺/VE-cadherin⁺ tubular structures resembling vasculature at both hair cycle stages analyzed (Fig. 4g, h). However, to our surprise these structures never stained positive for mature EC markers CD31, CD34, and Lyve1 (Fig. 4i–k) in >45 tdTomato⁺ tubular structures/staining/stage analyzed. Thus, our in vivo lineage tracing analysis did not support the computational prediction of *Cdh5*⁺/*Krt19*⁺ cells as a potential EVP subpopulation that may contribute to mature vasculature formation in adult skin during the hair cycle. This left unresolved the identity and function of the *Cdh5*⁺/*Krt19*⁺ population and of the tubular structures it formed.

## *Cdh5*⁺/*Krt19*⁺ cells encapsulate neurofilament-positive nerve bundles

Recently, scRNA-seq of cells isolated from sciatic nerves were profiled and a *Krt19*⁺ population was reported as a putative perineurial population[37,38], although in vivo validation was lacking. Perineurial cells are a specialized non-EC type that encases thick nerve bundles, contributing to the nerve basement membrane and to nerve structural integrity[39].

Our *Cdh5*⁺/*Krt19*⁺ population molecular signature overlapped to some extent with the reported *Krt19*⁺ perineurial population (Fig. 5a–c), including some known perineurial genes such as *Slc2a1*, *Perp* and *Itgb4*[40] (Fig. 5b). VE-cadherin expression was not previously reported in these *Krt19*⁺ cells, but a decades-old histological study reported VE-cadherin staining around peripheral nerves[41]. We then stained skin sections from *Krt19*-CreERT; tdTomato mice with nerve-specific neurofilament (NF) and schwann cell-specific (S100β) antibodies. The data demonstrated exclusive localization of the tdTomato⁺ tubular structures with skin nerve bundles, which they appeared to encapsulate, as demonstrated by high-resolution confocal microscopy (Fig. 5d, e). Perineurial cells secrete basement membrane related proteins[42,43] that also matched with the strong ECM signature of the *Cdh5*⁺/*Krt19*⁺ population (Supplementary Fig. 6g). In fact, the *Krt19*-CreERT; tdTomato marked structures stained strongly for laminin, a basement membrane marker (Fig. 5f). We had already shown that the *Krt19*-CreERT; tdTomato structures expressed the VE-cadherin protein (Fig. 4g, h). In addition, *Cdh5*-CreERT²; tdTomato marking by TM injection at PD17 followed by NF staining also revealed abundant colocalization of tdTomato⁺ tubular structures around the nerves (Fig. 5g), confirming our findings. Collectively, these data demonstrate the identity of our *Cdh5*⁺/*Krt19*⁺ cell population as a previously unrecognized non-endothelial VE-cadherin⁺ perineural cell population that encapsulates the skin nerve bundles being embedded in basement membrane depositions (Fig. 5h).

## Dynamic remodeling of VE-cadherin⁺ perineurial cells and nerve bundles in hair cycle

Since VE-cadherin expression in ECs underlines highly dynamic tubular vascular structures that are strongly remodeled during hair cycle (Fig. 1a, b)[15,16,18], we wondered if our detection of VE-cadherin in the perineurial tubes may also underline dynamic structures. This may be especially the case, since skin nerves reportedly remodel during the hair cycle, when the hair follicle (HF) epithelium and the nerves likely crosstalk[15,44,45] (Fig. 6a).

Nerve bundles arrive in skin through the subcutaneous muscle and form 3 horizontal plexuses, with the middle plexus at the hypodermal/dermal junction; medium-sized and thin nerve bundles derived from this plexus innervate the bulge, upper hair follicle and the epidermal site[15,44,45] (Fig. 6a). During anagen, the hypodermal nerves become stretched out and display new nerve terminations innervating the newly formed hair follicle (HF bulbs), while dermal nerves display increased density of thin filaments and more branching around the HF bulge[44,45]. The early reports documenting nerve remodeling during hair cycle[44,45] were consistent with our images of thick (70 μm) NF/VE-cadherin-stained skin sections, followed by z-stack confocal imaging at telogen and anagen (Fig. 6b-d'). Dynamic remodeling of the nerves was also apparent in our quantification, where thick bundles were more frequently scored at telogen than at anagen (Fig. 6e).

To see if the VE-cadherin⁺ perineurial structures also undergo remodeling during hair cycle, we used *Krt19*-CreERT; tdTomato mice TM induced at PD17 and analyzed skin at telogen (PD20) and anagen (PD32) for changes in non-HF tdTomato⁺ structure morphology (Fig. 6f–i). At telogen, the tdTomato⁺ structures outside the HF appeared compact and only rarely extended into long tubular structures, which appeared more prominent and frequent by anagen (Fig. 6g, h). The tdTomato⁺ signal enclosed variable structures that contained 1–12 total DAPI⁺ nuclei within the nerve bundle. Quantification of the number of nuclei belonging to schwann cells that were enclosed by tdTomato signal/structure demonstrates an average increase and a distribution shift towards larger structures at anagen (Fig. 6i). This data may suggest that perineurial cells spread or stretch out around and over the nerves and schwann cells. Ki67 staining of skin sections at the different stages did not catch co-localization with the perineurial cells, suggesting little if any proliferation of these cells.

To document how VE-cadherin⁺ perineurial cells associate with different nerve bundles during the hair cycle, we performed co-immunostaining with NF and VE-cadherin antibodies and divided the NF⁺ bundles into thick (>5 fibers), medium (3–5 fibers) and thin (1–2 fibers) (Fig. 6j). Quantification of this data at 3 hair cycle stages (*n* = 3 mice/stage and *N* > 45 bundles/stage) showed ~100% VE-cadherin⁺ perineurial signal associated with the thick bundles, ~40–80% association with the medium bundles depending on hair cycle stage, and only ~10% association with thin bundles or individual axons (Fig. 6k).

Taken together, these data demonstrate a dynamic homeostatic re-organization of VE-cadherin⁺ non-endothelial perineurial cells along with the skin nerve bundles and the hair follicles during hair cycle and a preference of the perineurial cells for thick nerve bundles (Fig. 6l).

## Alk1 regulates expression of ECM-associated genes in VE-cadherin (*Cdh5*)⁺ lineages

To understand the function and regulation of the VE-cadherin⁺ non-endothelial perineurial population during hair cycle, we turned to known VE-cadherin interacting partners from ECs[3,46]. Of these, the TGF-β pathway surfaced most strongly in our skin EC populations during hair cycle (Fig. 2h). In fact, we have reported one TGF-β receptor - activin receptor-like kinase 1 (Alk1) known as an important regulator of EC function[47,48]- as playing an essential role in skin vasculature remodeling during hair cycle[16,18]. Here, we found that *Cdh5*⁺/*Krt19*⁺ cells indeed expressed detectable levels of several TGF-β members, including Alk1 (Fig. 7a). To ask if Alk1 may play a role in the *Cdh5*⁺ perineurial cells, we first examined our previous scRNA-seq dataset generated from tdTomato⁺ sorted cells from *Cdh5*-CreERT² x Alk1^fl/fl mice TM induced at PD17 and sacrificed at 1ˢᵗ telogen[16], which we referred to from here on as KO (Fig. 7b–e). When compared with the telogen dataset that served here as wild-type control (CT), hundreds of genes changed expression in *Cdh5*⁺ perineurial cells due to Alk1 loss, and of these, 53 changed by >3x, which is highly significant in single-cell analysis (Supplementary Table 1). Among the 33 downregulated genes, we noted signaling processes, cytoskeletal, and ECM and basement membrane formation (Supplementary Table 1 and Supplementary Data 6). Notably, expression of encapsulating structure signatures, ECM constituents, and collagen-associated ECM, which were the top functional categories enriched in the *Cdh5*⁺ perineurial cells,

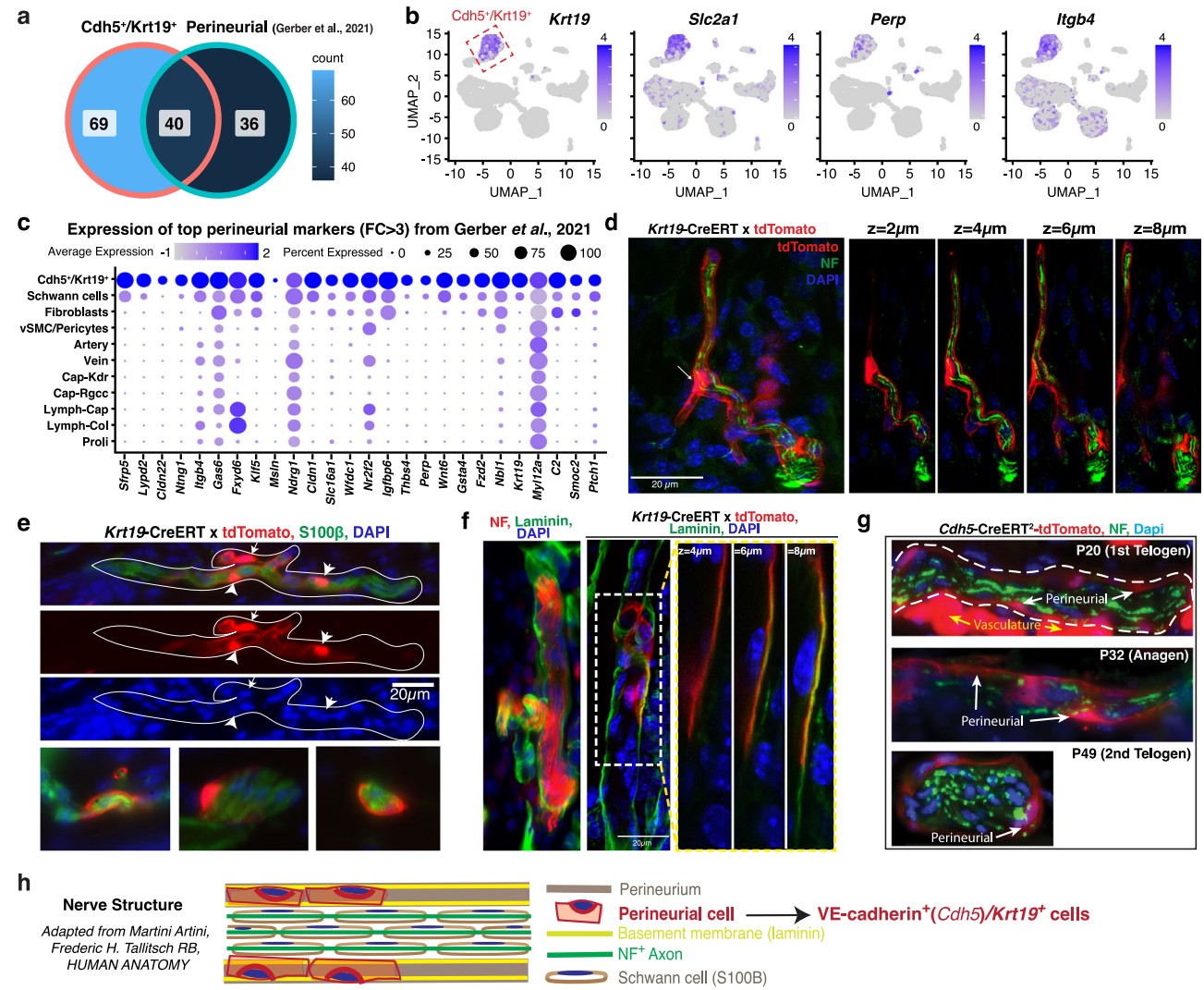

**Fig. 5 | *Cdh5*⁺/*Krt19*⁺ cells are non-endothelial perineurial cells embedded in the nerve basement membrane. a** Venn diagram showing overlap of gene signatures from the *Cdh5*⁺/*Krt19*⁺ population and predicted perineurial population from Gerber et al.[37]. Source data are provided as a Source Data file. **b** Feature plots for selected perineurial genes *Krt19, Slc2a1, Perp* and *Itgb4* from Carr et al.[40]. **c** Dotplot showing expression of top perineurial genes from Gerber et al.[37] in all populations (Immune populations were removed for less complicated representation in this panel). **d** High-resolution confocal optical Z-stacks maximum projection and individual optical planes showing encapsulation of neurofilament (NF, green) positive nerve fibers by *Krt19*-CreERT;tdTomato (red) positive cells. Tamoxifen was injected at PD17 and staining was performed at PD32 (anagen). Arrow points to tdTomato⁺ cell body that shows strong signal inside and around the nucleus from which long cytoplasmic extensions spread out around the nerve bundle forming nonvascular tubular structures. Scale bar 20 μm. (*n* = 3 biologically independent samples). **e** Co-immunofluorescence staining of S100β (green) on *Krt19*-CreERT;tdTomato (red) mice skin sections confirming presence of *Krt19*⁺ cells around schwann cells bundle. Arrows point to the cell body from which long cytoplasmic extension emanate wrapping around the nerves. (*n* = 3 biologically independent samples). **f** Left: IF image showing laminin (green) expression around NF⁺ (red) bundles. Right: Confocal Z-stacks showing co-localization of *Krt19*-CreERT; tdTomato (red) positive cells with basement membrane component laminin (green) positive cells outside the hair follicle (in dermis). Scale bar 20 μm. (*n* = 3 biologically independent samples). **g** *Cdh5*-CreERT²;tdTomato (red) skin sections stained for NF⁺ nerve bundles (green). (*n* = 3 biologically independent samples). **h** Cartoon depicting organization of peripheral nerve bundle (fascicle) assembly (Adapted from Martini Artini, Frederic H. Tallitsch RB, Human Anatomy).

were also significantly reduced in KO perineurial cells (Fig. 7f, g). To compare the Alk1 effects in ECs with those in the *Cdh5*⁺ perineurial cells, we used GO terms analysis using DEGs (FC > 1.3). Interestingly, several top down-regulated processes in *Cdh5*⁺ perineurial cells showed similarities in blood vessels (but not in lymphatic vessels) EC populations (Fig. 7g), which includes collagen-containing ECM, external encapsulating structure signatures, and oxidative metabolism. To further investigate if Alk1 has direct roles in regulating expression of these ECM constituents, we FACS sorted tdTomato⁺ cells from the skin tissue of *Cdh5*-CreERT² x Alk1^fl/fl x tdTomato mice only after 3 days of tamoxifen injection (Supplementary Fig. 7a). We then qPCR profiled mRNA levels in CT and KO cells that confirmed downregulation of ECM

factors *Col1a1, Lypd2, Col3a1, Serpinh1, Pcolce, Col1a2, Sbspon, Vtn* and *Fn1*, which were the top genes downregulated by Alk1 loss in perineurial cells (Fig. 7h).

Taken together, these findings demonstrate that Alk1, a known VE-cadherin partner in ECs function[3,46–48], is also utilized by the *Cdh5*⁺ non-endothelial perineurial cells to regulate gene expression signatures, defining both common and unique characteristics of cells within the skin VE-cadherin lineage. In perineurial cells, we find that Alk1 specifically promotes expression of ECM molecules along with the encapsulating structure identity. This was likely associated with the known function of this cell type in basement membrane deposition around the nerves[39,42].

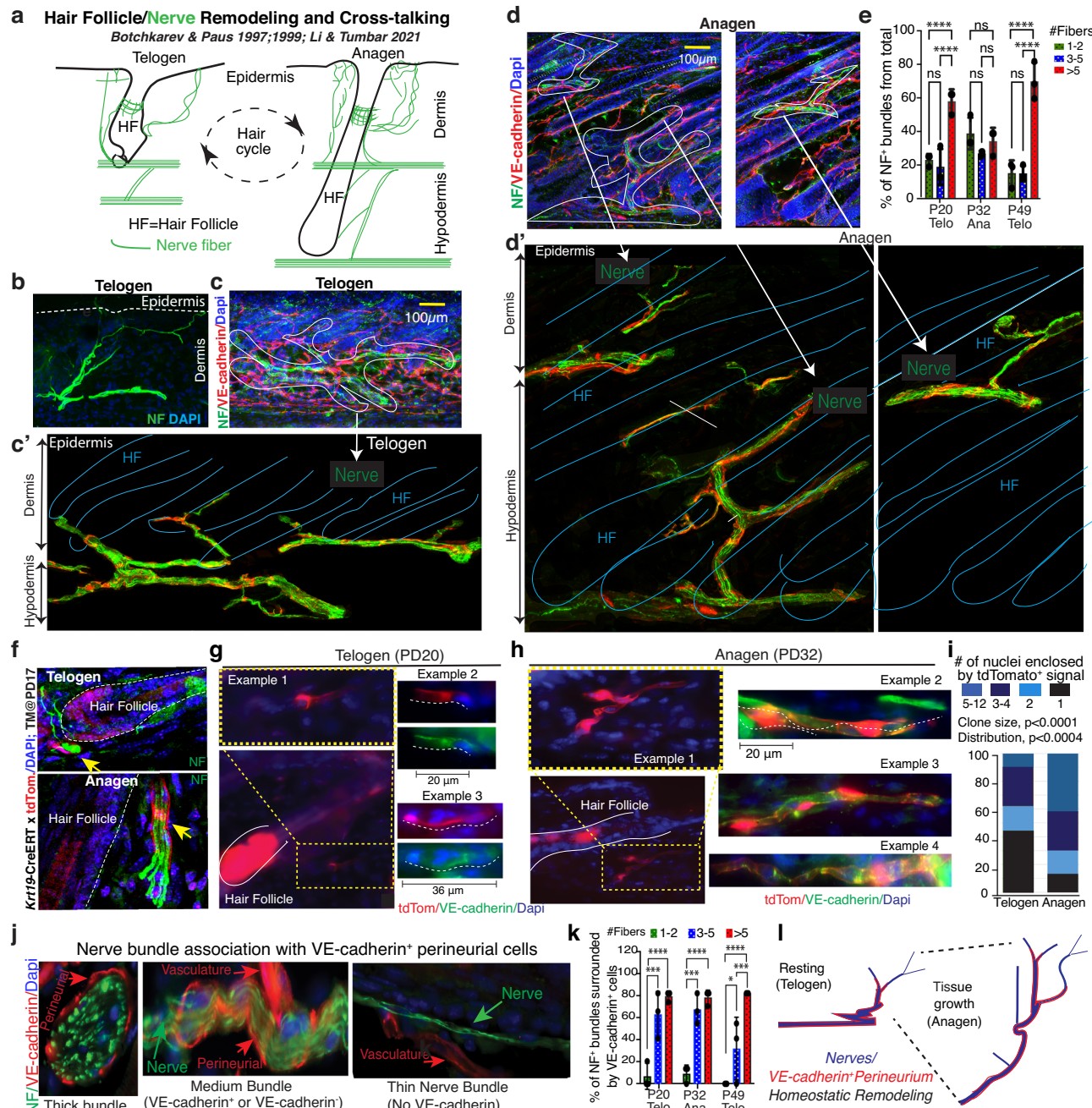

**Fig. 6 | Homeostatic remodeling of perineurial VE-cadherin⁺ lineage during the hair cycle. a** Cartoon depicting nerve remodeling during hair follicle (HF) growth from telogen to anagen, as previously described in Botchkarev & Paus 1997;1999[44,45]. Cross-talking between the two compartments was recently summarized[15]. **b** NF staining displaying tree-like branched nerve structures in telogen skin. **c-d'** Maximal projections of z-stack confocal images from 70 μm thick skin tissue sections stained with NF (green), VE-cadherin (red) and Dapi (blue) at telogen (**c**) and anagen (**d**). The enlarged images in (**c'**) and (**d'**) were digitally modified from their original in (**c**) and (**d**), respectively, for qualitative visualization of large complex nerve fibers within thick tissue samples. Visible NF⁺ bundles were manually traced and the surrounding tissue structures and all Dapi signal were digitally removed for visualization purposes. The HFs were pre-traced in the Dapi channel (blue lines). Scale bar 100 μm. **e** Quantification of NF⁺ nerve bundles showing distributions of #fibers/structure at telogen and anagen. n = 3 mice/stage, Data are presented as mean ± SD. Two-way ANOVA followed by a Tukey's multiple comparisons was used. ns - not significant, ****p ≤ 0.0001. Source data are provided as a Source Data file. **f** Confocal imaging of neurofilament (NF,

green) immunofluorescence staining on skin sections from *Krt19*-CreERT;tdTomato (red) mice. Tamoxifen (TM) was injected at PD17 and staining was performed at PD20 (telogen) or PD32 (anagen). **g, h** Lineage tracing of *Cdh5⁺/Krt19⁺* perineurial cells using *Krt19*-CreERT; tdTomato mice from (**f**). Tubular structures outside hair follicles appeared more elongated and branched at PD32. **i** Quantification of numbers of nuclei enclosed by tdTomato positive cells at telogen and anagen, as shown in panels (**g**) and (**h**). Statistics for clone size was derived using an unpaired two-tailed t test and for clone distribution using K-S test. Source data are provided as a Source Data file. **j** IF images of tissue sections immunostained for NF (green), VE-cadherin (red) and Dapi (blue). **k** Quantification of NF positive nerve bundles (like those in **j**) and their association with VE-cadherin⁺ cells (% from more than 48 fibers at each stage), n = 3 mice/stage, Data are presented as mean ± SD. Two-way ANOVA followed by a Tukey's multiple comparisons was used. ns - not significant, *p = 0.027, ***p ≤ 0.0008, ****p ≤ 0.0001. Source data are provided as a Source Data file. **l** Cartoon summarizing remodeling and straightening of nerve bundles during skin tissue expansion at anagen.

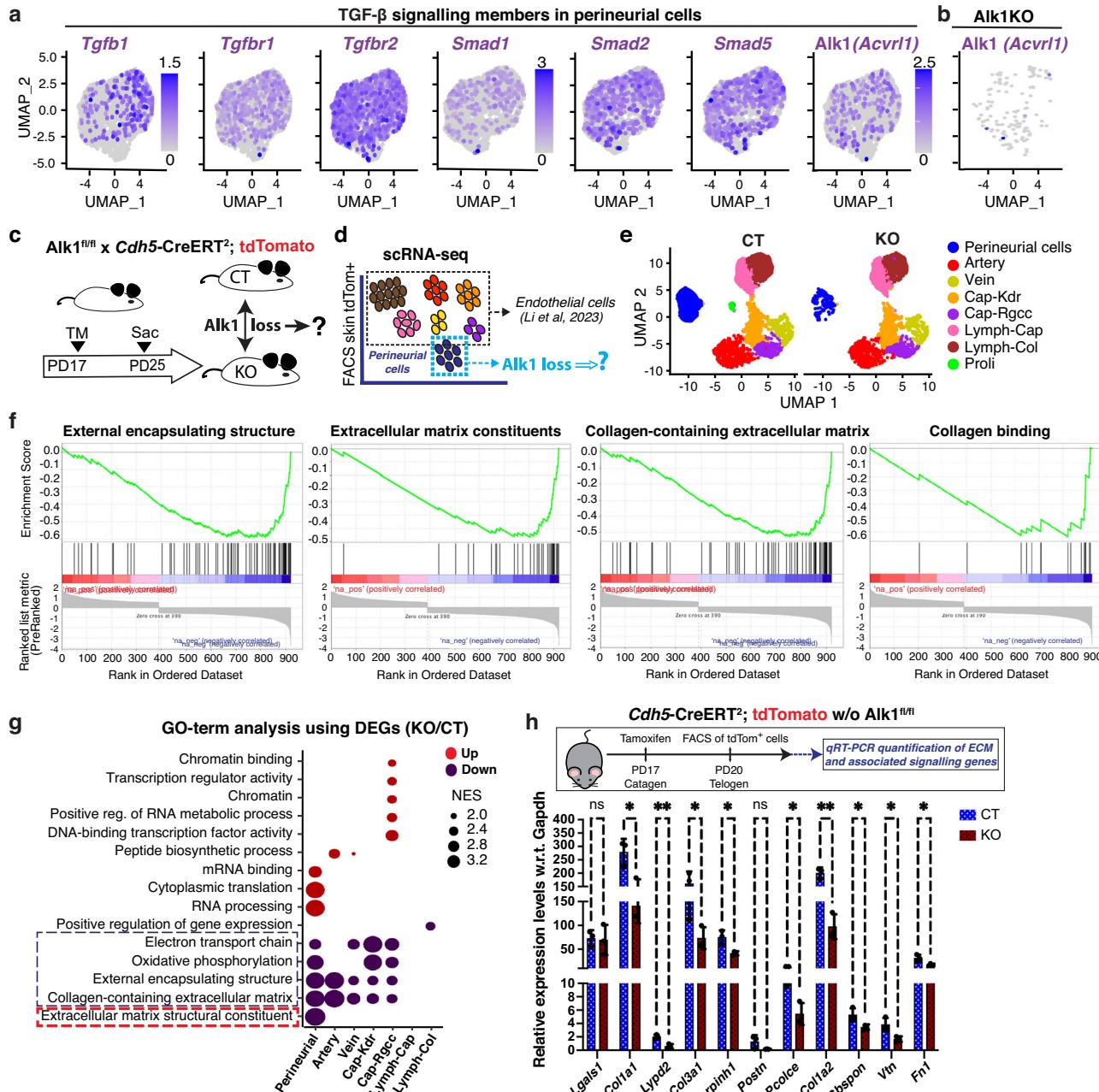

**Fig. 7 | Alk1 regulates expression of encapsulation and basement membrane components in vivo. a** Feature plots displaying expression of TGF-β signaling associated genes including Alk1 in *Cdh5*⁺/*Krt19*⁺ perineurial cells. **b** Feature plot of Alk1 mRNA (*Acvrl1*) level in *Cdh5*⁺/*Krt19*⁺ perineurial cells from KO mice. **c, d** Schematic depicting workflow for isolation and analysis of *Cdh5*⁺/*Krt19*⁺ perineurial cells and ECs from Alk1^fl/fl^x *Cdh5*-CreERT²; tdTomato mice[16]. **e** UMAP clustering showing all ECs and perineurial cells in wild-type (CT) and KO mice. **f** GSEA analysis showing striking depletion of ECM, encapsulation and collagen binding signatures

in perineurial cells of KO mice. **g** GO-term analysis shows a comparison of pathways/functions changed by Alk1 loss in perineurial cells with EC populations. Source data are provided as a Source Data file. **h** Real-time PCR validation of ECM-associated down-regulated genes in perineurial cells of KO mice. Tamoxifen was injected at PD17 and tdTomato⁺ cells were FACS sorted at PD20 to investigate direct effect of Alk1 deletion on ECM genes expression. $n = 3$ mice/genotype. Data are represented as mean ± SD. Multiple unpaired t tests was used. ns - not significant, *$p \leq 0.042$, **$p \leq 0.0061$. Source data are provided as a Source Data file.

## Alk1 acts in perineurial cells to maintain proper nerve branching during the hair-cycle

To understand how Alk1 loss in perineurial cells may affect nerve organization and remodeling, we first examined skin from *Cdh5*-CreERT² x Alk1^fl/fl^; tdTomato mice, TM induced at PD17 and sacrificed at PD25, by NF and laminin staining. Little if any changes were noted relative to control mice at this early stage post Alk1 loss. Since these mice die shortly after PD25 due to defects in the vasculature, we crossed the *Krt19*-CreERT x Alk1^fl/fl^; tdTomato mice and examined

tdTomato⁺ clones outside the HFs at PD32 and PD49 after TM induction at PD17. These mice survived and appeared normal 2-3 weeks post TM induction with no obvious phenotypes, indicating that loss of Alk1 in *Krt19*-CreERT targeted cells does not induce major physiological effects in these adult mice. This may not be surprising as the patches of non-HF tdTomato⁺ cells are relatively small and rare with only 1-3 structure present in each 30−60 μm thick and ~1.5 cm long skin sections analyzed. Second, since *Krt19*-CreERT also targets Alk1 in the hair follicles, where it is lowly expressed (Supplementary Fig. 7b), we

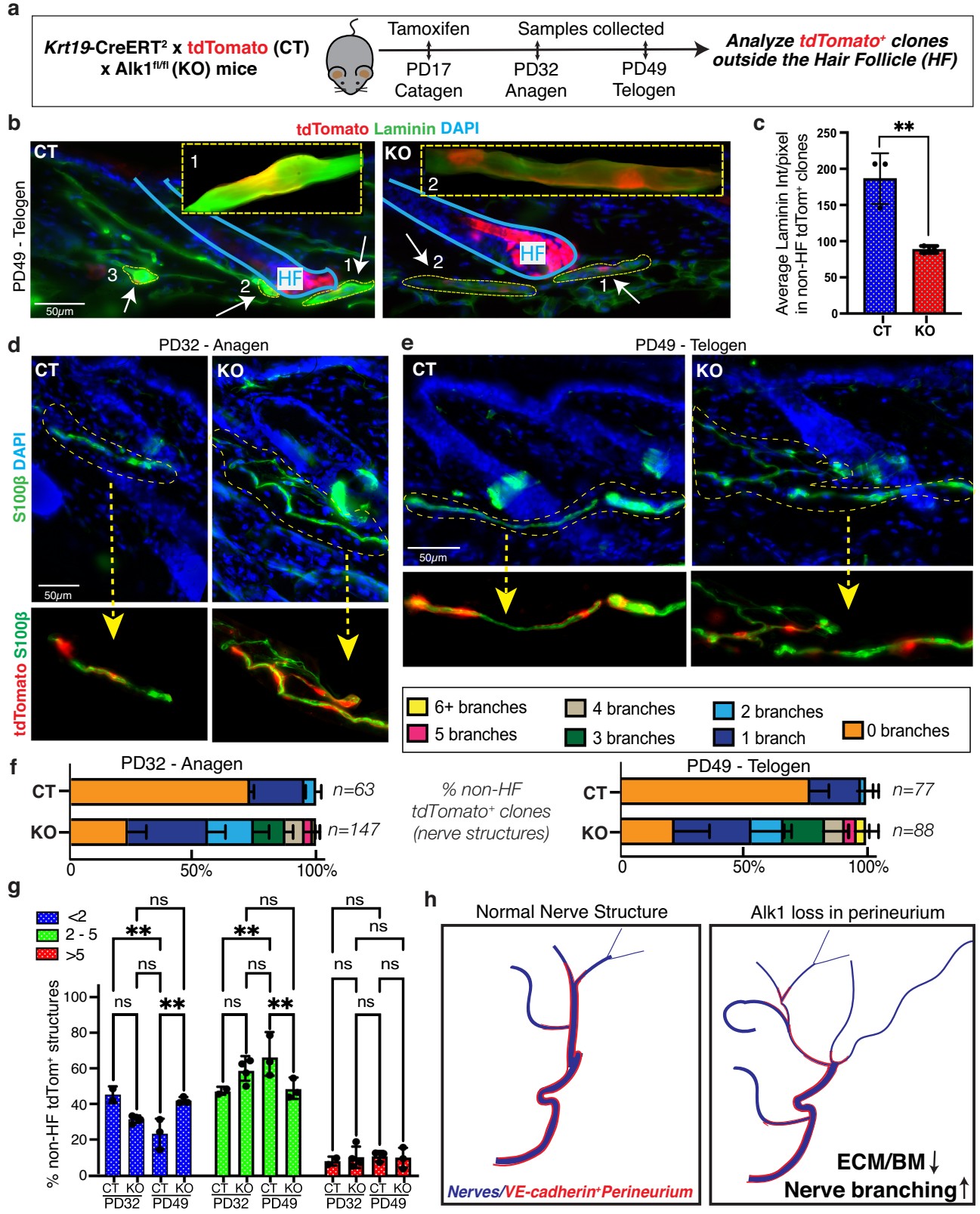

**a** *Krt19*-CreERT² x tdTomato (CT) x Alk1^fl/fl (KO) mice. Tamoxifen PD17 Catagen; Samples collected PD32 Anagen; PD49 Telogen. *Analyze tdTomato⁺ clones outside the Hair Follicle (HF)*

**b** tdTomato Laminin DAPI. CT / KO PD49 - Telogen. HF.

**c** Average Laminin Int/pixel in non-HF tdTom⁺ clones. CT vs KO. **

**d** PD32 - Anagen. CT / KO. S100β DAPI / tdTomato S100β.

**e** PD49 - Telogen. CT / KO.

6+ branches / 5 branches / 4 branches / 3 branches / 2 branches / 1 branch / 0 branches

**f** PD32 - Anagen. CT n=63, KO n=147. % non-HF tdTomato⁺ clones (nerve structures). PD49 - Telogen. CT n=77, KO n=88.

**g** <2 / 2 - 5 / >5. % non-HF tdTom⁺ structures. CT KO PD32 / CT KO PD49.

**h** Normal Nerve Structure / Alk1 loss in perineurium. *Nerves/VE-cadherin⁺Perineurium*. ECM/BM↓ Nerve branching↑

inspected the tissue for any defects in hair morphology or hair cycle progression. However, these appeared normal at both PD32 (anagen) and PD49 (telogen) stages analyzed in all *n* = 3 mice/stage analyzed (Supplementary Fig. 7c). The tdTomato⁺ non-HF cells and clones were present at their normal relatively rare frequency of 1–3 nerve structures/skin section in both CT and KO skin. Interestingly, IF staining for

laminin – a marker that provides a collective assessment of the basement membrane's architecture and stability in nerves and other structures - was strongly reduced in *Krt19*-CreERT targeted tdTomato⁺ clones in dermis of KO when compared with CT mice (Fig. 8a-c).

Next, we examined nerve bundle morphology in the tdTomato⁺ non-HF clones using IF-staining for S100β, a schwann cell marker that

**Fig. 8 | Alk1 loss in perineurial cells results in increased nerve branching during hair-cycle. a** Schematic of workflow for assessment of effect of long-term Alk1 loss in *Krt19*-tdTomato[+] clones outside hair follicles (HFs) in CT and KO mice. *Krt19*-CreERT driver was used for simultaneous tdTomato labeling and Alk1 deletion in targeted clones. **b** IF image showing laminin (green) coated tdTomato[+] (red) perineurial cell clones outside HFs, and decreased laminin intensity in KO clones. Scale bar 50 μm. **c** Quantification of laminin staining intensity per *Krt19*-tdTomato[+] clone in 12 μm thin skin tissue sections of CT and KO. Per pixel signal value for each clone from *n* = 3 mice/genotype was measured using Fiji and plotted using prism 9. Data are presented as mean ± SD. An unpaired two-tailed t test was used. **p = 0.0087. Source data are provided as a Source Data file. **d, e** IF image of S100β along with *Krt19*-tdTomato showing increased splitting and perturbed organization of nerves in KO structures at P32 anagen (**d**) and P49 telogen (**e**). Scale bar 50 μm.

**f** Quantification of number of branch points per continuous nerve structure in the dermis region. The relative fraction (%) of each nerve category was compared between genotypes and plotted using bar plots. *N* > 60 nerve structures per genotype were analyzed. Data are presented as mean ± SD. Source data are provided as a Source Data file. **g** Comparison of nerve bundles containing different thickness (# of fibers) structures between CT and KO at anagen and telogen, n = 3 mice for CT and *n* = 4 mice for KO. Data are presented as mean ± SD. Two-way ANOVA followed by a Tukey's multiple comparisons was used. ns - not significant, **p ≤ 0.0077. Source data are provided as a Source Data file. **h** Cartoon summarizing implications of Alk1 function in the perineurial cells in preventing excessive nerve branching during tissue expansion, potentially through maintaining expression of basement membrane ECM proteins.

shows a tightly colocalized staining pattern with NF (Supplementary Fig. 7d). Strikingly, nerves marked as tdTomato[+] clones appeared disorganized, highly branched and with fewer fibers/bundle in *Krt19*-CreERT driven Alk1 KO skin dermis as compared to the CT dermis at both hair cycle stages analyzed (Fig. 8d, e). This was largely confirmed by our quantification, especially in the number of branches per tdTomato[+] nerve structure in CT and KO mice in more than 60 structures for each genotype/stage (Fig. 8f, g). Together, these results identified Alk1-driven molecular control of perineurial function and established one role of these perineurial cells in maintaining the physiological nerve branching patterns during the hair cycle via Alk1 activity (Fig. 8h).

## Discussion

Our work re-defines the cellular and molecular heterogeneity of the essential VE-cadherin[+] lineage and its dynamics during homeostasis, firmly expanding its functions beyond classical endothelial vasculature remodeling[1,49] into perineurial cells and skin nerve maintenance. Most importantly, we uncovered a previously unrecognized VE-cadherin[+] non-endothelial perineurial cell population encapsulating the skin nerves, contributing to the nerve basement membrane, and remodeling into tubular-like structures along with the nerves during the hair cycle. We implicate Alk1, a known VE-cadherin interacting partner from the TGF-β pathway, in the function of these perineurial cells. We show that Alk1 in the perineurium promotes production of ECM factors, deposition of nerve basement membrane, and maintenance of proper branching and organization of nerve bundles during the hair cycle.

Importantly, VE-cadherin expression in a non-endothelial (perineurial) population and the implication of its endothelial partner Alk1[3,46–48], in homeostatic remodeling of nerve structures, changes the existing dogma in the field that considers these interacting factors as specific to the endothelium. Our result is supported by a decades-old, and since largely ignored VE-cadherin staining near skin nerves[41]. VE-cadherin is a cell-cell contact molecule with specialized function in tubular structures that have been so-far thought of as exclusively vascular, until now. VE-cadherin is known to confer tubular EC structures flexibility, resilience, and permeability to gases, nutrients, and blood/immune cells[50–52], and it'll be interesting in the future to examine its role in the perineurium. Given its known functions in endothelial cells, the identification of VE-cadherin in the non-endothelial dynamic perineurial tubes is intriguing. The perineurium is already known to be endowed with characteristics that VE-cadherin promotes such as being flexible, resistant to mechanical damage, and playing important roles in permeability through the blood-nerve barrier[53]. In fact, acute and chronic pain is related to nerve inflammation and the infiltration of immune cells to the nerve terminations[54], which may be mediated in the perineurium via VE-cadherin, a tantalizing possibility for future investigation.

Our data clearly implicated Alk1 in promoting ECM gene expression in both endothelial and perineurial cells, which both form dynamic VE-cadherin[+] tubular structures. On the other hand, Alk1

represses ECM gene expression in other non-endothelial mesenchymal cell types including structural but non-tubule forming cells, such as fibroblasts[55] and chondrocytes[56]. This suggest that although perineurial cells have a strong mesenchymal identity, Alk1's effect on ECM genes in perineurial cells resembles more the endothelial, than the mesenchymal cell types. Interestingly, Alk1 loss in endothelial cells results in increased permeability and leakiness of the vasculature, which loses its integrity[57]. Similarly, we show that nerve bundles branch more when Alk1 is lost from their surrounding perineurium, possibly due to low laminin content in the basement membrane. In contrast, the known Alk1 role in EC migration[16] may not apply in perineurial cells, since the latter seem to remodel correctly into branched out tubular structures at anagen. This remodeling may involve perineurial cell stretching, migrating and encapsulating schwann cells and nerve terminations, and this seems to occur even in the absence of Alk1. Interestingly, gene expression changes upon Alk1 loss are more similar in perineurial and blood vessel endothelial cells, when compared to lymphatic vessel ECs, perhaps attesting to the less dynamic nature of the latter.

That the VE-cadherin/Alk1 signaling axis, previously implicated exclusively in the endothelium[3,46–48], is present in the perineurial cells as well, may have important implications in understanding the known coordinated nerve/vasculature remodeling that has intrigued scientists for decades[58]. Nerve and vasculature structures are created by branching morphogenesis, and both structures share and respond to growth factor derived from the other compartment[58]. Moreover, they are both known to respond to signals from hair follicle epithelial cells during the hair cycle[15]. Perhaps a VE-cadherin[+]/Alk1 axis may be part of a general tissue network that drives the coordinated nerve/vasculature remodelling during homeostasis, with implications in vasculature and nerve disease.

In addition to shifting the dogma for the VE-cadherin[+] population by expanding its known heterogeneity, our data provides the first high resolution cellular and molecular characterization of vasculature dynamics in the homeostasis of a highly regenerative tissue. While many adult vascular systems are quiescent during homeostasis[59], a few adult tissues besides skin[15] also employ dynamic vasculature changes in the absence of injury. The latter include menstruation in female endometrial homeostasis[60], exercise in the skeletal muscle[61], and the ongoing regeneration of lacteals in the intestine[62]. The synchronicity of hair follicle homeostatic growth and regression in skin facilitates the characterization of this system, illuminating the way for other adult regenerative tissues. Interestingly, we detected an anagen-specific increase in blood ECs at the expense of the lymphatic ECs, suggesting the two lineages are in opposite flux during skin homeostasis. While blood vessel EC number increase is not surprising, and aligns with proposed high energy needs to supply $O_2$ and nutrients for the production of new hair shafts[15,20] and to expand the skin hypodermis, the concomitant decrease in lymphatics ECs is intriguing. A reported reduction in the dermal lymphatic capillaries area at anagen[24], and the reduced drainage observed at anagen[23] may corroborate our finding.

The lymphatic vasculature is generally static in adult tissues[63,64], but homeostatic regeneration also occurs in the intestinal 'lacteal' lymphatic capillaries and the ovary during folliculogenesis[62,65]. Our study in skin adds important insight into active vascular homeostasis of another highly regenerative adult tissue.

Importantly, we exclude here the existence of a hair-cycle active skin resident VE-cadherin[+] endovascular progenitor (EVP) that was previously documented in the aorta, skin wounding and skin tumors[11–13]. Our ultra-sensitive labeling method (*Cdh5*-CreERT² x tdTomato) uncovered a mesenchymal perineurial cell type and not an immature EC type. Thus, it appears that skin ECs change their overall numbers and balance between blood and lymphatic ECs during hair cycle, without cellular input from a skin-resident VE-cadherin[+] immature cell type or EVP.

Furthermore, our data demonstrate that in homeostasis, skin EC populations coordinately change their molecular makeup to reflect increased metabolic and biosynthetic activities. This corroborates with previous findings of microvasculature remodeling through capillary pruning and duplication[66] and through exercise-induced angiogenic stimulus[61]. Several major signaling pathways surfaced from our transcriptomic analysis as differentially regulated in the specific EC populations, including TGF-β, Notch and Wnt signaling. These data should aid in the emerging cross-communication of ECs with hair follicle stem cells[15], which is poorly understood. More importantly, this work may aid in efforts to engineer proper organotypic skin cultures for chronic and diabetic wounds[67] and for understanding skin vascular disease such as Human hereditary hemorrhagic telangiectasia (HHT)[68].

Finally, our study has some technical strengths and advances worth mentioning. First, using FACS on the highly sensitive *Cdh5*-CreERT²;tdTomato labeling system permitted access to rare and low-expressing VE-cadherin cell types, which baring one decades-old study[41] were generally missed in antibody staining. Second, despite our sensitive methods, we did not find a VE-cadherin[+] endothelial vascular progenitor (EVP), as suggested in the aorta, in cancer or in injury[11–13]. Significantly, our computational lineage trajectory analysis and specific gene expression signatures as done in aorta[11] supported the EVP hypothesis but this was refuted by our genetic lineage tracing in skin, cautioning against the use of single-cell transcriptomic computational methods alone to predict cell identity and functions. Third, our single-cell transcriptomic data revealed strong expression of *Krt19* mRNA in perineurial cells, which prompted us to employ the *Krt19*-CreERT driver that also marked the skin perineurium. The *Krt19*-CreERT was previously considered specific to epithelial hair follicle stem cells[36], but evidently, this assumption should be revised given our data. The expression of a keratin in a perineurial cell type with a strong mesenchymal signature is intriguing and requires further investigation and confirmation at the protein level.

In conclusion, our systematic analysis of the VE-cadherin[+] population in adult skin revealed previously un-recognized non-endothelial cell types and identified dynamic reorganization that highlights the coordinated nerve and vasculature remodeling during homeostasis. This data un-earths unexpected cellular and functional heterogeneity in the essential VE-cadherin[+] population using skin as a highly regenerative adult tissue and will have relevance to other adult tissues and to vascular and nerve disease.

## Methods
### Mice and treatments
All the mouse work was performed in accordance with the Cornell University Institutional Animal Care and Use Committee (IACUC) guidelines (protocol no. 2007-0125). The tdTomato[26] (Stock no 007905) and *Krt19*-CreERT[36] (Stock no 026925) mice were obtained from Jackson Laboratory. The Alk1[fl/fl] and *Cdh5*-CreERT²[25] mice were imported from Dr. Anne Eichmann at Yale University and used as previously described[18]. Mouse sex was not considered in the study

design. The mice were intraperitoneally (IP) injected with tamoxifen at a dose of 200 μg/g body weight to induce Cre recombination. For BrdU-pulse experiments, mice were fed with 0.8 mg/ml BrdU in daily water supply.

### Immunofluorescence staining, microscopy and image processing
Immunofluorescence (IF) staining on the non-prefixed tissue sections was performed by following the standard protocol, as described previously[18]. The tdTomato[+] skin tissues were prefixed in 4% paraformaldehyde for 2 h at 4 °C followed by passing them through a sucrose gradient before embedding in OCT compound (Tissue Tek, Sakura). The non-prefixed samples were first fixed in 4% paraformaldehyde (PFA) for 10 minutes at room temperature (RT), followed by washing with PBS and 20 mM glycine in PBS. The sections were blocked with 5% normal serum for 1 h at RT before incubating them with primary antibodies at 4 °C overnight. Next day, sections were 3x washed with PBST and then incubated with secondary antibodies (1:500). The sections were washed 3x with PBST and stained with DAPI. The slides were mounted with antifade solution. For thick sections (70 μm), primary antibodies incubation was performed for 48 hours at 4 °C and secondary antibodies overnight at 4 °C. For BrdU staining, tissue sections or cells were first stained with non-BrdU primary antibodies, followed by fixation using 4% PFA. DNA was denatured by incubating the slides in 1 M HCl solution for 55 minutes at 37 °C. The slides were washed with PBS and re-blocked using 0.5% Tween-20 and 1% BSA before incubating them with anti-BrdU antibody overnight at 4 °C. Then slides were washed in PBST and incubated with FITC-conjugated anti-rat antibody. Primary antibodies used in this study include CD31 (1:100, BD Biosciences, 550274); VE-cadherin (1:500, R&D Systems, AF1002); LYVE1 (1:400, Thermo Scientific, 14-0443-82); Prox1 (1:300, Abcam, ab199359); BrdU (1:300, Abcam, ab6326); Endomucin (1:300, Santa Cruz Biotechnology, sc-65495); CD34 (1:100, BD Biosciences, 553731); Laminin (α1) (1:500, Sigma-Aldrich, L9393); S100β (1:400, Proteintech, 15146-1-AP) and Neurofilament Heavy chain (1:500, Millipore, AB5539). The IF images were captured using either the Leica DMI6000B microscope with Leica K5 camera or the Zeiss LSM inverted 880 confocal microscopes. The z-stacks (1 μm stacks) images of thick sections were deconvoluted using AutoQuant X software and were further processed for brightness, contrast and stitching using Fiji[69]. Multiple images were combined using stitching package[70] in Fiji[69] for Fig. 1. The quantification graphs were generated using GraphPad Prism 9.

### Fluorescence-activated cell sorting (FACS) of VE-cadherin[+] cells
FACS isolation of VE-cadherin[+] cells and its purity check was performed as previously described[27]. Briefly, tdTomato (Jax Stock #007905), *Cdh5*-CreERT²[25] and *Krt14*-H2BGFP[71] mice were used for VE-cadherin[+] cells and control keratinocytes isolation. The VE-cadherin[+] cells were labeled with tdTomato by injecting tamoxifen (200 μg/g) at postnatal day (PD)17, and dorsal skin was collected at PD20 or PD32. The dorsal skin was digested in collagenase and Dispase mixture as previously described[27]. The dead cells were removed by LIVE/DEAD™ Fixable Aqua Dead Cell Stain Kit (ThermoFisher). FACS Aria (BD Biosciences) was used for cell sorting in Cornell Flow Cytometry facility. FACS data were analyzed with the FlowJo (FlowJo™ Software, v10.5.0, BD Biosciences).

### Single cell library preparation and data analysis
The barcoded single-cell 3′ cDNA libraries were generated using Chromium Single Cell 3′ gel bead and library Kit v3 (10x Genomics) and sequenced using an Illumina NextSeq-500. The raw data were aligned to the mouse reference genome (mm10-2020-A) using the 10X Genomics *Cell Ranger* pipeline (v6.0.1). The data analysis was performed in R using the Seurat package version 4.0[72,73]. Cells that had between 200

and 5000 genes expressed and had under 10% of the UMIs mapped to mitochondrial genes were retained. We obtained a total of 3957 and 8627 high-quality cells for telogen and anagen samples, respectively. For further analysis, all samples were merged, the transcript counts were log-normalized, and the expression of each gene was scaled so that the variance in gene expression across cells was one, followed by Principal Component Analysis (PCA). The data integration was performed by PCA embeddings using *Harmony*[74]. The corrplot R function was used to measure the correlation between the two biological replicates. Differentially expressed genes (DEGs) were identified by '*FindAllMarkers*' function using the Wilcox Rank Sum test.

### Cell lineage trajectory, gene scoring and gene ontology analysis
Trajectory analysis was performed using Monocle[35]. We used the top 2000 genes to order cells in pseudotime trajectory by Monocle and the cells having lower than 200 transcripts were removed from further analysis. The DDRTree package was employed to construct the trajectories. To determine the collective enrichment of a set of genes in different populations, the AddModuleScore function in the Seurat R package was used. For pathway analysis, DEGs were identified by Seurat's '*FindMarkers*' function using the Wilcox Rank Sum test (FC > 1.3). Further enrichment analysis and plotting were performed as previously described[27].

### Heatmap of differentially expressed genes (DEGs) using raw counts
First, DEGs were identified using *FindMarkers* function of *Seurat*. Next, to obtain the expression value for each gene in all populations, the raw counts for the DEGs were extracted using the *FetchData* function. Of 205 genes (FC > 1.5) 6 showed 0 values in at least one population, whereas the corresponding population in the other stage showed a very low expression level relative to other populations. Thus '0' was replaced with the low 'background' value from the corresponding population at the other stage. This way, these genes with very low expression levels in some populations appear 'unchanged' from telogen to anagen, thus eliminating the possibility that small fluctuations in background expression are assigned a 'changed' value. The *Xist* gene showed 0 values in more than one population but did not display an obvious background value in the corresponding population at the other stage, and thus they were not analyzed further. The same method was applied to the gene set of 784 genes (FC > 1.3), where 18 showed 0 values in at least one population.

### RNA isolation and real-time quantitative RT-PCR
FACS sorting of VE-cadherin[+] cells was performed as described above. Sorted cells were directly collected in TRIzol LS (ThermoFisher) for total RNA isolation, and c-DNA synthesis was performed using Superscript III (Invitrogen) following manufactures instructions. The primer sequences are provided in Supplementary Data 7.

### Statistics and reproducibility
Data comparisons between two groups was performed using the two-tailed Student's t-test. For multiple comparisons, we used two-way ANOVA followed by Tukey's test. We used GraphPad Prism (Prism 9) for statistical calculations and analyses. Statistical significance was indicated in all experiments by *p* value. No statistical method was used to predetermine the sample size. The experiments were not randomized. The investigators were not blinded to allocation during experiments and outcome assessment.

### Reporting summary
Further information on research design is available in the Nature Portfolio Reporting Summary linked to this article.

## Data availability
The scRNA-seq data generated in this study have been deposited in the NCBI Gene Expression Omnibus (GEO) database under the accession number GSE211381. Source data are provided with this paper.

## Code availability
R script code for Seurat analysis is deposited in a GitHub repository and is freely accessible at https://github.com/GChovatiya/Chovatiya_et_al_Nat_Comm_SourceCode.

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

## Acknowledgements

We thank Drs. Lauren D. Walter and Benjamin D. Cosgrove for their help with scRNA-seq analysis and helpful suggestions. We also thank the Center for Animal Resources and Education (CARE) for mouse care and the Cornell University Biotechnology Resource Center (BRC) for their help in scRNA-seq experiments. Confocal microscopy work on Zeiss upright 880 was supported by the Cornell BRC facility (NYSTEM C029155 and NIH S10OD018516). This work was supported by NIH/NIAMS Grants R01AR070157, R56AR081021, R01AR081021 and R01AR073806 to TT and by the Empire State Stem Cell Fund through New York State Department of Health Contract # C30293GG training grant to GC. Opinions expressed here are solely those of the author and do not necessarily reflect those of the Empire State Stem Cell Board, the New York State Department of Health, or the State of New York.

## Author contributions

G.C., N.L. and T.T. designed the experiments. G.C., N.L. and J.L. performed the experiments. G.C. did the computational analysis and in vivo validations. N.L. did the clonal analysis in Fig. 6. J.L. performed laminin and S100β staining and their quantifications in Fig. 8. S.G. contributed to pathway analysis in Fig. 2. G.C. and T.T. analyzed and interpreted all data. G.C., N.L. and T.T. prepared the figures. G.C. and T.T. wrote the manuscript.

## Competing interests

The authors declare no competing interests.
