## [Peer Review File · Nature Communications]

Alk1 acts in non-endothelial VE-cadherin+ perineurial cells to maintain nerve branching during hair homeostasisREVIEWER COMMENTS

Reviewer #1 (Remarks to the Author):

This manuscript by Chovatiya et al. reports the identification of a novel Cdh5+/Krt19+ skin perineural cell population and its transcriptional and lineage dynamics in the mouse hair cycle. scRNA-seq analysis of Cdh5-CreER-labelled skin cells show that there is a Cdh5+/Krt19+ non-endothelial cell population in telogen and anagen dorsal skin, in addition to other Cdh5+ blood and lymphatic cell populations. This Cdh5+/Krt19+ population encapsulates skin nerve fibers and shares a transcriptional signature with previously reported perineurial cells. Krt19-CreER lineage tracing shows that they increase colony size within perineurial structure during hair cycle, but do not contribute to blood and lymphatic vessel lineages in normal skin homeostasis, despite their transcriptional similarity to previously reported endothelial vascular progenitors (EVPs). Moreover, deletion of the VE-cadherin interacting partner Alk1 downregulates signature genes of Cdh5+/Krt19+ cells, suggesting its involvement in the identity of Cdh5+/Krt19+ cells.

Overall, the authors provide high quality data and careful analysis on this interesting cell type. However, the manuscript provides relatively shallow descriptive data with weak coherence. More importantly, the functional importance of this cell population has not been addressed, leaving its contribution in tissue homeostasis and regeneration unclear. Therefore, I would have to say the level of conceptual advance and broad interest of this study is relatively minor and does not seem to reach the level required by this journal. I have a few comments below for the authors to consider, to further enhance the impact of the work.

1. This paper is composed of three low relevant stories; 1) characterization of the gene expression changes in endothelial cell populations, 2) Alk1 knockout and 3) lineage analysis of Cdh5+/Krt19+ cell population. They are not framed in a coherent story. This is the major problem that makes this paper relatively superficial. One of the possible avenues to strengthen this paper would be to add functional analysis of this cell type.
2. The title of the paper is ambiguous. More direct and specific words and sentence should be used. The abstract also needs more clarity and logical flow.
3. The detection of differentially expressed genes between telogen and anagen vessel subpopulations is to be commended and could be a useful database for future studies, but it does not provide significant technical and conceptual advances in the present study.

Reviewer #2 (Remarks to the Author):

In this manuscript, authors combine single-cell profiling with Cre-lox based lineage marking to define types of skin cells that express VE-cadherin (VE-cad). They posit that populations of VE-cad+ cells are dynamic in skin even under normal conditions, changing significantly with stages of the growth activity by skin's follicles. The latter seems to be a historically established fact (i.e. that skin vasculature remodels as skin's follicles cycle), but using single-cell approach, authors now provide more "resolution" to this process (i.e. they define and examine several vascular and lymphatic VE-cad+ populations). Beyond this, authors show that while studying VE-cad+ cells, they found one sizable population that also expresses keratin 19 gene and that does not seem to be an endothelial cell type. Further profiling of these cells points to their perineurial identity. These VE-cad+/Krt19+ cells seem to form tube-like sheaths around large and medium size skin nerves, where their role can be in depositing basement membrane type of extracellular matrix.

Technically, the study seems to be of a good quality. Single-cell and histological analyses both appear to be done diligently. Lineage marking experiments are helpful in validating some of the single-cell data predictions, mainly the peri-neurial identity of VE-cad+ cells. At the same time I have the following concerns:

- 1) While quite detailed, the study seems mostly of a descriptive nature. Authors classify several endothelial cell types, all of which would be expected to exist in most vascularised tissues, such as skin. The follicle growth associated remodelling of these endothelial populations seems to match what has been reported both in classic studies and in several recent papers that authors cite. So, new data here just strengthens the point that skin with active follicles is more vascularised, but its an incremental addition of knowledge.

Further, identification of peri-neurial non-endothelial VE-cad+/Krt19+ is a good addition to the story, albeit it is too, descriptive. In this work authors stop short of saying what these new cells do and for example, what would happen to skin nerves (or skin in general) if they are removed (such as genetically?) or made more abundant

(sorted and injected?).

2) Related to the point above, authors show some single-cell data apparently from another manuscript under consideration (Fig. 4e), where it seems that these peri-neural VE-cad⁺ cells become depleted in Alk1^{fl/fl} x Cdh5-CreERT2 mutant mice. Do these mice have skin nerve defects?

Also, addition of unpublished data from one manuscript to another seems unconventional. There must be a more conventional way of doing it. Such as, first paper can be deposited to bioRxiv for review?

3) From the nerve biology point of view, it is unclear just how significant is the finding that basement membrane making perineural cells can express VE-cad and Krt19? Perhaps, these genes are just marker genes and are not functionally essential? It seems that testing this can be possible by deleting VE-cad from Krt19-CreERT mice, which authors have. Also, it is unclear if Krt19 is made at the protein level and what can a keratin filament do in such mesenchymal cells.

Generally, study will be more exciting if it is supported by functional experiments, revealing role of newly identified peri-neural cell type in the skin.

Reviewer #3 (Remarks to the Author):

In this manuscript, the author reports a new non-endothelial cell cluster which is defined by Cdh5+Krt19+. This type of cell is distributed around the skin nerves. The authors suggest that the Cdh5+Krt19+ cells belong to EVP subpopulation that can contribute to blood vessels during the hair cycle, but are immature blood vessels. Further, they find that during the hair cycle, this group of nerve-enveloping cells can also be involved in maintaining the homeostasis of the skin nerves. This is an interesting finding and will improve our understanding of VE-cadherin expressing subpopulation involved in hair cycling and nerve remodeling. However, to make the article more rigorous, some questions need to be addressed.

1. The authors report a novel Cdh5+Krt19+ subpopulation cells for maintain skin homeostasis. However, the functional study of this group of cells in vivo is not enough. The authors should do specific clearance experiments of Cdh5+Krt19+ population to verify whether they have effect on hair cycle or nerve remodeling.

2. The authors use VE-cadherin staining and Cdh5-CreER;tdTomato mice to observe the endothelial cells in HPuHG. But the results presented by the two methods appear to be inconsistent. The density of endothelial cells in the HypoDe region is higher than that in the De region in figure 1b, but the proportions seem to be opposite in supplementary figure 1a. Please explain the difference in the results of the two methods.

3. The authors demonstrate that endothelial cells have proliferated by Brdu staining of Cdh5-CreER;tdTomato mice, but this does not mean that the Cdh5+Krt19+ subpopulation has also proliferated. The authors also suggested that this cell population belong to EVP that can contribute to the vessel. Can the authors please give evidence that this subpopulation can proliferate.

4. In figure 4a, the authors show that Alk1 is highly expressed in the Cdh5+Krt19+ subpopulation, but Tgfb2 and Smad2/5 shown in Figure 4a seem to be more expressed. Are Tgfb2 and Smad2/5 involved in these subpopulation regulation?

5. Alk1 is expressed in all endothelium of the skin, and the authors use Cdh5-CreER; Alk1 fl/fl to detect its functional regulation in the Cdh5+Krt19+ subpopulation. Do the knockout experiment cause Cdh5+Krt19- cells conversion to Cdh5+Krt19+ subpopulation, which may affect the results of the experimental analysis? How could the author rule out this effect?

6. Are Krt19-CreER mice specific for cell labeling? Will Cdh5+ mature vessel be marked? In figure 5g, the authors use Krt19-CreER to trace Cdh5+Krt19+ cells and prove that they contribute to tubular vasculature, but the resolution of VE-cadherin antibody staining is relatively low and prone to false positives. The authors need to confirm this conclusion by using higher resolution methods, such as genetic methods.

7. The authors used two mice, Krt19-CreER and Cdh5-CreER, respectively, to demonstrate that the Cdh5+Krt19+ subpopulation is distributed around the nerves. However, there is no direct lineage tracing evidence for Cdh5+Krt19+ subpopulation. The authors need to provide direct evidence to prove the distribution and functions of this cell population, for example, use other combined recombination reporters.

8. The authors find that Cdh5+Krt19+ subpopulation does not contribute to mature blood vessels at steady state. Could this subpopulation contribute to mature vessels during after skin injury?

9. In figure 5d, the title should be "...change..."

Section 1. Common Major Revisions:

We thank all 3 reviewers for their time and effort to review our manuscript, highlight the strengths and quality of our data and analyses, and provide constructive criticisms and suggestions. The reviewers point out that identification of a novel non-endothelial VE-cadherin (also known as *Cdh5*) expressing population of perineurial cells in skin is the most novel aspect of our study. In the previous version, we showed that these VE-cadherin⁺ perineurial cells encapsulate the skin nerves and are remodeled along with the nerves during the hair cycle. These perineurial cells expressed a large variety of ECM and basement membrane molecules, which may aid in the structural role of perineurial cells in maintaining nerve integrity. The reviewers all requested some sort of perturbations in this perineurial population to investigate its function during the hair cycle, and to improve on what was viewed as the descriptive nature of our previous study.

We considered the 3 suggested strategies to perturb the VE-cadherin⁺ (*Cdh5*⁺) perineurial cells, by either: 1) cell depletion, 2) cell enrichment, or 3) genetic modification. The goal would be to then observe possible effects on either hair cycle and nerve remodeling to elucidate the function of these cells in skin.

1) Cell depletion: We have attempted to kill the VE-cadherin⁺ perineurial cells by crossing the *Cdh5*-CreERT² x Rosa-26-DTA mice, locally inducing cell death with 4-hydroxytamoxifen (4-OHT) by topical application in a small skin area. This spotted induction should avoid severe secondary systemic effects from massively killing the vasculature along with the perineurial cells if we inject the tamoxifen. DTA induction was followed by hair plucking to induce hair cycle and nerve remodeling. Although we observed some active Caspase-3 positive cells near the nerves, suggesting killing of a few perineurial cells, most nerves remained enmeshed by these structures, and were unperturbed in their morphology. Given perineurial cells seemed resistant to killing and given our 3rd strategy (genetic modification) gave us very exciting results, we did not pursue this strategy further.

2) Cell enrichment: A suggestion was made that we transplant the sorted perineurial cells in the skin to observe effect on nerves and hair cycle. We briefly considered this possibility but given the high stress the cells would be subjected to upon isolation and the unlikely situation that transplanting would neatly reconstruct the 3D nerve structures they likely form gradually during skin development, we decided to invest our efforts in genetic modification of perineurial cells to examine their function in skin.

3) Genetic modification: A salient function of perineurial cells is in depositing basement membrane around nerve bundles providing mechanical protection and maintaining nerve integrity^{1,2}. Alk1 is a known VE-cadherin interacting partner from endothelial cells functioning as a receptor in the TGF- β pathway³. We have already demonstrated that Alk1 is important in endothelial cells for skin vasculature remodeling during hair cycle^{4, 5}, but its role in nerve

remodeling via perineurial cells has not been reported to our knowledge. In the previous version of this paper, we showed by scRNA-seq that *Alk1* loss in the perineurial cell cluster resulted in massive downregulation of the ‘encapsulating structure’ genes and ECM/ basement membrane genes (Fig 7f-h). Here we confirmed this finding for a handful of genes by performing qRT-PCR of FACS-sorted tdTomato⁺ cells from *Cdh5*-CreERT²; tdTomato⁺; *Alk1*fl/fl mice just 3 days after TM injection (Fig. 7i).

Analysis of tens of images for multiple cohorts of mice at PD25 (TM injected at PD17) left us empty handed as nerve bundles in both CT and KO skin samples were intact, as exemplified below.

Unfortunately, the *Cdh5*-CreERT² x *Alk1*fl/fl mice die within 10 days of TM induction and display a delayed telogen phase in part due to vasculature defects⁴. Hence, in our previous submission, we were unable to study the long-term effect of *Alk1*

loss in perineurial cells in nerve organization during hair cycle. To circumvent this problem, in the current revised paper, we induced the *Alk1* loss in rare clones of perineurial cells using the *Krt19*-CreERT; tdTomato x *Alk1*fl/fl mice. We then examined the hair cycle and the nerve phenotypes in the tdTomato⁺ clones at 15 days (anagen PD32) and 32 days (PD49) upon TM induction. *Krt19*-CreERT; *Alk1*KO mice survived the treatment well and displayed a normal hair cycle and similar abundance, size, and localization of the tdTomato⁺ perineurial clones in the *Alk1*KO vs CT.

Excitingly, the tdTomato⁺ Cre targeted perineurial clones were abnormal as they strongly downregulated laminin deposition in the basement membrane around nerve bundles in *Alk1*KO mice when compared with CT. This was accompanied by striking increased nerve branching and frequency of thin nerve bundles in *Alk1*KO tdTomato⁺ clones at both full anagen (PD32) and second telogen (PD49) (Fig 8). Therefore, we conclude that at least one function of our VE-cadherin⁺ perineurial cells is in maintaining proper branching of the skin nerve bundles as they are remodeled during the hair cycle. In conclusion, we uncover a novel role of *Alk1* - a known partner of VE-cadherin in endothelial cells – in perineurial cells where it promotes proper nerve organization and branching, most likely through enforcing the expression of basement membrane-associated genes (Figure 8f, g).

We believe that with the new data above, we have now addressed the major concerns of the reviewers, making our study more suitable for publication in Nature Communications. Below we address in more detail each reviewer’s concerns point by point.

Section 2. Point-by-point response to reviewers' comments

Reviewer #1 (Remarks to the Author):

This manuscript by Chovatiya et al. reports the identification of a novel Cdh5+/Krt19+ skin perineural cell population and its transcriptional and lineage dynamics in the mouse hair cycle. scRNA-seq analysis of Cdh5-CreER-labelled skin cells show that there is a Cdh5+/Krt19+ non-endothelial cell population in telogen and anagen dorsal skin, in addition to other Cdh5+ blood and lymphatic cell populations. This Cdh5+/Krt19+ population encapsulates skin nerve fibers and shares a transcriptional signature with previously reported perineurial cells. Krt19-CreER lineage tracing shows that they increase colony size within perineurial structure during hair cycle, but do not contribute to blood and lymphatic vessel lineages in normal skin homeostasis, despite their transcriptional similarity to previously reported endothelial vascular progenitors (EVPs). Moreover, deletion of the VE-cadherin interacting partner Alk1 downregulates signature genes of Cdh5+/Krt19+ cells, suggesting its involvement in the identity of Cdh5+/Krt19+ cells.

Overall, the authors provide high quality data and careful analysis on this interesting cell type. However, the manuscript provides relatively shallow descriptive data with weak coherence. More importantly, the functional importance of this cell population has not been addressed, leaving its contribution in tissue homeostasis and regeneration unclear. Therefore, I would have to say the level of conceptual advance and broad interest of this study is relatively minor and does not seem to reach the level required by this journal. I have a few comments below for the authors to consider, to further enhance the impact of the work.

- 1. This paper is composed of three low relevant stories; 1) characterization of the gene expression changes in endothelial cell populations, 2) Alk1 knockout and 3) lineage analysis of Cdh5+/Krt19+ cell population. They are not framed in a coherent story. This is the major problem that makes this paper relatively superficial. One of the possible avenues to strengthen this paper would be to add functional analysis of this cell type.*
- 2. The title of the paper is ambiguous. More direct and specific words and sentence should be used. The abstract also needs more clarity and logical flow.*
- 3. The detection of differentially expressed genes between telogen and anagen vessel subpopulations is to be commended and could be a useful database for future studies, but it does not provide significant technical and conceptual advances in the present study.*

We agree that the identification of VE-cadherin expression in a non-endothelial lineage highlighting perineurial cells that remodel during hair cycle is the most novel and exciting part of our work. As described at the top of this response, we have now added long-term analysis of Alk1 loss in patches of perineurial cells demonstrating a role of the perineurial cells in maintaining proper nerve branching during hair cycle-associated remodeling (see **Section 1** above and Figures 7i and Figure 8 of the revised manuscript).

We hope the reviewer agrees that with the new additions of the long-term Alk1KO study, and our re-writing the paper significantly gained both more coherence and functional insight.

Our vision in the previous submission has been to present this paper as a thorough and comparative characterization of the dynamics, heterogeneity, and gene expression changes of the 'VE-cadherin lineage' using skin as a highly regenerative model tissue. VE-cadherin expression previously was thought to be entirely specific to endothelial cells, while we are now adding several classes of non-endothelial – including the interesting *Krt19*⁺ population - in the skin during hair cycle, hence the original broad title and the inclusion of genes and pathways for endothelial cells. We felt that the discovery of VE-cadherin is no longer just an endothelial specific factor was highly relevant, but we agree with the reviewer that more functional substance was needed for the new population we uncovered – so we added here the long-term Alk1KO data as described above.

In terms of other novel points besides the perineurial cells, not many tissues besides skin undergo vasculature and nerve remodeling in homeostasis (in the absence of injury or tumorigenesis). A thorough characterization of heterogeneity and gene expression changes of VE-cadherin⁺ endothelial cells, cell type by cell type, during homeostatic remodeling is lacking, which makes our pathways interesting to highlight especially for the vasculature audience. Furthermore, several studies included in Nature Communications uncovered a VE-cadherin⁺ endovascular progenitor cell in skin wounds and in melanoma ^{6, 7, 8}. Our data demonstrate that in normal skin homeostasis a VE-cadherin⁺ EVP was not present, providing additional advance in the field. Finally, while we could retain the changes in signaling pathways of endothelial cells for a different study, we feel that it is important in this story to emphasize the overall dynamic nature of all skin VE-cadherin expressing lineages during hair cycle, and for direct comparison of endothelial with the novel perineurial cells, which both strengthens our findings of the new population, and broadens the relevance of our study beyond skin and hair cycle into vasculature and nerve remodeling during homeostasis. Alk1 function in the remodeling of both skin nerves and skin vasculature now unifies to some extent the regulation of the distinct skin VE-cadherin⁺ lineages, increasing coherence of the story. We tried to convey these ideas and new data and findings by connecting the dots more in the paper and in the discussion section. Moreover, in the revised manuscript, we have also modified the title and abstract to make it more specific, as requested by this reviewer, and to include the new findings on the function of perineurial cells in nerve maintenance. Once again, we thank the reviewer for the useful pointers that improved our manuscript.

Reviewer #2 (Remarks to the Author):

In this manuscript, authors combine single-cell profiling with Cre-lox based lineage marking to define types of skin cells that express VE-cadherin (VE-cad). They posit that populations of VE-cad⁺ cells are dynamic in skin even under normal conditions, changing significantly with stages of the growth activity by skin's follicles. The latter seems to be a historically established fact (i.e. that skin vasculature remodels as skin's follicles cycle), but using single-cell approach, authors

now provide more "resolution" to this process (i.e. they define and examine several vascular and lymphatic VE-cad⁺ populations). Beyond this, authors show that while studying VE-cad⁺ cells, they found one sizable population that also expresses keratin 19 gene and that does not seem to be an endothelial cell type. Further profiling of these cells points to their perineural identity. These VE-cad⁺/Krt19⁺ cells seem to form tube-like sheaths around large and medium size skin nerves, where their role can be in depositing basement membrane type of extracellular matrix.

Technically, the study seems to be of a good quality. Single-cell and histological analyses both appear to be done diligently. Lineage marking experiments are helpful in validating some of the single-cell data predictions, mainly the peri-neural identity of VE-cad⁺ cells. At the same time, I have the following concerns:

1) While quite detailed, the study seems mostly of a descriptive nature. Authors classify several endothelial cell types, all of which would be expected to exist in most vascularised tissues, such as skin. The follicle growth associated remodelling of these endothelial populations seems to match what has been reported both in classic studies and in several recent papers that authors cite. So, new data here just strengthens the point that skin with active follicles is more vascularised, but its an incremental addition of knowledge.

Although the skin vasculature has been shown to physically remodel during the hair cycle, changes in the specific cell type fraction and molecular identity, if any, was unknown. For example, while lymphatic vessels were known to remodel morphologically, it was not known that their cell number decreases during hair cycle. Our single-cell analysis provides molecular and cellular resolution highlighting the unusual and unique dynamics of skin vasculature during normal homeostasis. Furthermore, studies in the aorta, skin wounds and tumors uncovered the existence of a VE-cadherin⁺ endovascular progenitor cell^{6,7,8}, and it was possible that this EVP progenitor also exists and acts during the hair cycle; we believed our data ruled out this important model.

Further, identification of peri-neural non-endothelial VE-cad⁺/Krt19⁺ is a good addition to the story, albeit it is too, descriptive. In this work authors stop short of saying what these new cells do and for example, what would happen to skin nerves (or skin in general) if they are removed (such as genetically?) or made more abundant (sorted and injected?).

Please see **Section 1** of this response to reviewer for a thorough summary of how we addressed this point. Just briefly here, we now demonstrate a role of these perineurial cells via Alk1 in maintaining proper nerve branching during hair cycle-associated remodeling (see Section 1 above and Figures 7i and Figure 8 of the revised manuscript).

2) Related to the point above, authors show some single-cell data apparently from another manuscript under consideration (Fig. 4e), where it seems that these peri-neural VE-cad⁺ cells become depleted in Alk1^{fl/fl} x Cdh5-CreERT2 mutant mice. Do these mice have skin nerve defects?

The perineurial cells do not become depleted in the Alk1KO mice, the change noted by the reviewer is due to fractional increase in the number of endothelial and immune cells that come down in the KO FACS sort. Regarding effects of Alk1 loss in perineurial cells on skin nerves, this is an excellent question that we believe we fully addressed it in **Section 1 ‘Common Major Revisions’** above. Briefly, because the *Cdh5-CreERT²;Alk1 fl/fl* mice die before the hair follicles enter the hair cycle, we were unable to address the effect of Alk1 loss in perineurial cells during hair cycle. We have now circumvented this problem by utilizing the *Krt19-CreERT* driver that can remove Alk1 from patches of perineurial cells. We demonstrate that nerve bundles branch more and show increased single fibers during hair cycle, when Alk1 is lost from the perineurial cells (please see **Section 1 ‘Common Major Revisions’** above.) and new Figure 7i and 8.

Also, addition of unpublished data from one manuscript to another seems unconventional. There must be a more conventional way of doing it. Such as, first paper can be deposited to bioRxiv for review?

The data from this manuscript is now fully published and cited correctly in the paper, and here is a link to it: <https://www.embopress.org/doi/full/10.15252/embj.2022112196>.

3) From the nerve biology point of view, it is unclear just how significant is the finding that basement membrane making perineurial cells can express VE-cad and Krt19? Perhaps, these genes are just marker genes and are not functionally essential? It seems that testing this can be possible by deleting VE-cad from Krt19-CreERT mice, which authors have. Also, it is unclear if Krt19 is made at the protein level and what can a keratin filament do in such mesenchymal cells.

The Reviewer has raised an interesting question here, and we do hope to address in future the role of VE-cadherin in perineurial cells. However, since we do not currently have access to the VE-cadherin fl/fl mice, this is a long-term endeavor that we cannot address in a timely manner here. Based on the known roles of VE-cadherin in vasculature, we can speculate that VE-cadherin may provide junctional flexibility and resilience^{9, 10, 11} which may be required during the nerve bundle remodeling in anagen. It may also, provide selective permeability to metabolites and immune cells^{12, 13} that are required for homeostatic maintenance of structures inside perineurium. We have discussed this part in the discussion section of the revised manuscript for future investigation.

Regarding *Krt19* expression in perineurial cells, despite our sustained attempts over the past few years, we could not find antibody working conditions, which left us wondering if the protein is really made in these cells. Moreover, the *Krt19* fl/fl mice are not available. So, in this case, we do regard *Krt19* as more of a helpful marker. Expression of keratins in endothelial cells is not entirely novel^{14, 15, 16}, although its function remains a mystery to date to our knowledge.

Generally, study will be more exciting if it is supported by functional experiments, revealing role of newly identified peri-neural cell type in the skin.

We fully agree with this point, and we spent the past 6 months thoroughly addressing it (please see answer to your point 1 and **Section 1 ‘Common Major Revisions’** above and new Figure 7i and 8). In short, we now provide evidence for the function of these perineurial cells in maintaining proper nerve branching during their remodeling in hair cycle and implicate Alk1 in the genetic control of this function, likely via control of expression of ECM and basement membrane genes.

Reviewer #3 (Remarks to the Author):

In this manuscript, the author reports a new non-endothelial cell cluster which is defined by Cdh5+Krt19+. This type of cell is distributed around the skin nerves. The authors suggests that the Cdh5+Krt19+ cells belong to EVP subpopulation that can contribute to blood vessels during the hair cycle, but are immature blood vessels.

We thank Reviewer 3 for reviewing our manuscript and providing comments. We want to clarify that although we probed the role of these novel VE-cadherin⁺ cells as a putative EVP, our data refuted that model. In fact, the Cdh5⁺/Krt19⁺ cells never contribute to vasculature formation mature or immature during the hair cycle. While we initially believed the tubular structures these cells formed to be immature blood vessels, we in fact found this to be a novel population of mesenchymal perineurial cells around the skin nerves. We apologize for the confusion, and we tried to clarify this data more in the text of the revised version.

Further, they find that during the hair cycle, this group of nerve-enveloping cells can also be involved in maintaining the homeostasis of the skin nerves. This is an interesting finding and will improve our understanding of VE-cadherin expressing subpopulation involved in hair cycling and nerve remodeling. However, to make the article more rigorous, some questions need to be addressed.

1. The authors reports a novel Cdh5+Krt19+ subpopulation cells for maintain skin homeostasis. However, the functional study of this group of cells in vivo is not enough. The authors should do specific clearance experiments of Cdh5+Krt19+ population to verify whether they have effect on hair cycle or nerve remodeling.

We fully agree with this point, also raised by the other reviewers and we spent the past 6 months thoroughly addressing it (please see **Section 1 ‘Common Major Revisions’** above and new Figure 7i and 8). In short, we now provide evidence for the function of these perineurial cells in maintaining nerve branching during hair cycle remodeling. We implicate Alk1, a VE-cadherin interacting partner³ and TGF-β receptor in genetic control of this perineurial cells likely via regulating expression of ECM and basement membrane genes.

2. The authors use VE-cadherin staining and Cdh5-CreER;tdTomato mice to observe the endothelial cells in HPuHG. But the results presented by the two methods appear to be

inconsistent. The density of endothelial cells in the HypoDe region is higher than that in the De region in figure 1b, but the proportions seem to be opposite in supplementary figure 1a. Please explain the difference in the results of the two methods.

In both images, we show that the vasculature is organized in form of horizontal plexus in the hypodermis region at telogen and remodels during anagen towards more vertical and dispersed structures, as we previously described^{4,5}. A relatively brighter signal in the dermis vs hypodermis region in the tdTomato images could be due to endogenous tdTomato fluorescent labeling as compared to VE-cadherin antibody staining in Figure 1. Also, tissue processing and staining procedure for both methods are very different (tdTomato tissues were prefixed using 4% PFA for 1hr and did not undergo standard IF staining procedure vs VE-cadherin sections were first cryo-sectioned and then fixed undergoing three days long IF staining procedure), which may introduce some variation in brightness of the stainings, specifically in those areas where the signal is weak like dermis as compared to the hypodermis.

3. The authors demonstrate that endothelial cells have proliferated by Brdu staining of Cdh5-CreER;tdTomato mice, but this does not mean that the Cdh5+Krt19+ subpopulation has also proliferated. The authors also suggested that this cell population belong to EVP that can contribute to the vessel. Can the authors please give evidence that this subpopulation can proliferate.

This unfortunately is part of the same confusion above, because although we tested the possibility that the *Cdh5*⁺/*Krt19*⁺ cells were EVPs, the lineage tracing results refuted that hypothesis. Our data demonstrate that this population is instead a specialized mesenchymal perineurial non-endothelial cell type. Moreover, while we have strong evidence from in situ staining corroborated with the FACS sorting and the scRNA-seq data that endothelial cells proliferate in hair cycle, we do not have good evidence nor do we claim that the perineurial cells proliferate. In fact, our IF staining at several stages could never identify Ki67 positive perineurial cells (see an example below). We believe the *Cdh5*⁺/*Krt19*⁺ perineurial cells do not divide much and are instead just physically remodeled during hair cycle stretching over the Schwann cells, which they encapsulate. We tried to clarify this somewhat more in the text. However, we feel that documenting the exact nature of the perineurial cell remodeling is a time costly endeavor and it would be best addressed in a different study with long pulses

and pulse-chase BrdU and live imaging perhaps combined in future with the VE-cadherin KO which could affect this remodeling.

4. In figure 4a, the authors show that Alk1 is highly expressed in the Cdh5+Krt19+ subpopulation, but Tgfbr2 and Smad2/5 shown in Figure 4a seem to be more expressed. Are Tgfbr2 and Smad2/5 involved in this subpopulation regulation?

As shown in Figure 7a of the revised manuscript, the *Cdh5*⁺/*Krt19*⁺ perineurial cells display expression of multiple factors involved in TGF- β signaling. We specifically focused on Alk1, a receptor of the pathway, for two reasons: its known interaction with VE-cadherin in endothelial cells and its role in vasculature remodeling in hair cycle. Thus, we felt Alk1 is a good candidate to test in non-endothelial VE-cadherin⁺ perineurial cells for roles in remodeling of nerves and of the perineurial cells (please see Section 1 ‘Common Major Revisions’ above and new Figure 7i and 8). The detailed investigation of other TGF- β members is out of scope for the current study, now that our new long-term Alk1KO in perineurial cells provided the missing functional data for the perineurial cells.

5. Alk1 is expressed in all endothelium of the skin, and the authors use Cdh5-CreER; Alk1 fl/fl to detect its functional regulation in the Cdh5+Krt19+ subpopulation. Do the knockout experiment cause Cdh5+Krt19- cells conversion to Cdh5+Krt19+ subpopulation, which may affect the results of the experimental analysis? How could the author rule out this effect?

Endothelial and perineurial cells traditionally have distinct origins and functions, and we did not see an obvious increase in *Cdh5*⁺/*Krt19*⁺ perineurial cells after Alk1KO using the *Cdh5*-CreERT² driver. Hence, based on current evidence, we do not think that Alk1 deletion in endothelial cells could lead to their conversion in *Cdh5*⁺/*Krt19*⁺ cells. Also, this concern is alleviated by our new data when we used the *Krt19*-CreERT driver to delete Alk1 in *Cdh5*⁺/*Krt19*⁺ cells, which is not active in *Cdh5*⁺/*Krt19*⁻ endothelial cells. The results suggest depletion of basement membranes and unraveling of the nerves, consistent with the previous Alk1 loss results obtained with the *Cdh5*-CreERT² driver.

6. Are Krt19-CreER mice specific for cell labeling? Will Cdh5+ mature vessel be marked? In figure 5g, the authors use Krt19-CreER to trace Cdh5+Krt19+ cells and prove that they contribute to tubular vasculature, but the resolution of VE-cadherin antibody staining is relatively low and prone to false positives. The authors need to confirm this conclusion by using higher resolution methods, such as genetic methods.

Our *Krt19*-CreER; tdTomato labeling strategy neither label *Cdh5*⁺ mature blood vessels (Figure 4i-k), nor these mature vessel cells express *Krt19* (Figure 1h). The *Krt19*-tdTomato structures outside the hair follicles label VE-cadherin⁺ cells outside the nerve bundles. While the expression of VE-cadherin in these cells is relatively weak, the junctional puncta appearance in high resolution

microscopy is very clear and indicates specificity. We invite the reviewer to check images in both Figure 4 and 5. We have confirmed our VE-cadherin antibody staining specificity and their co-localization with tdTomato signal in the same optical plane by capturing z-stacks images using the highest-resolution a modern confocal (i880) microscope can offer. We believe this data is extremely strong and are very confident with the interpretation.

7. The authors used two mice, Krt19-CreER and Cdh5-CreER, respectively, to demonstrate that the Cdh5+Krt19+ subpopulation is distributed around the nerves. However, there is no direct lineage tracing evidence for Cdh5+Krt19+ subpopulation. The authors need to provide direct evidence to prove the distribution and functions of this cell population, for example, use other combined recombination reporters.

We have thoroughly inquired the scRNA-seq data for possible specific or unique markers of this population. Unfortunately, while many genes were strongly upregulated in the *Cdh5*⁺/*Krt19*⁺ population relative to endothelial cells, all these genes were also expressed broadly in the skin fibroblasts, which would confuse the analysis even more. In the end, we decided to stick with the *Krt19*-CreERT driver, which although it targets the epithelial hair cells as well, it provides a clear view of the VE-cadherin⁺ perineurial cells outside the hair follicle. The only *Krt19*-tdTomato⁺ cells outside the hair follicle are the VE-cadherin⁺ perineurial cells, which supports our conclusions.

8. The authors find that Cdh5+Krt19+ subpopulation does not contribute to mature blood vessels at steady state. Could this subpopulation contribute to mature vessels during after skin injury?

This point is again unfortunately affected by the perception that the *Cdh5*⁺/*Krt19*⁺ population may function as an EVP, which is incorrect. This population is purely a perineurial mesenchymal non-endothelial cell type encasing thick nerve bundles. The perineurial cells are known to have a totally independent cellular identity and function from endothelial cells. So, it would be extremely unusual if during injury this population would transdifferentiate to endothelial cells and contribute to the formation of mature blood vessels. Moreover, the focus and strength of our paper is on vasculature and nerve remodeling during normal skin homeostasis. While the situation in injury is interesting, we believe will be best addressed in a proper different study.

9. In figure 5d, the title should be "...change..."

We thank the Reviewer for pointing out the typo. We have corrected the error in the revised Figure.

Reference:

1. Petrova ES, Kolos EA. Current Views on Perineurial Cells: Unique Origin, Structure, Functions. *Journal of Evolutionary Biochemistry and Physiology* **58**, 1-23 (2022).
2. Jaakkola S, Peltonen J, Uitto JJ. Perineurial cells coexpress genes encoding interstitial collagens and basement membrane zone components. *Journal of Cell Biology* **108**, 1157-1163 (1989).
3. Lagendijk AK, Hogan BM. Chapter Ten - VE-cadherin in Vascular Development: A Coordinator of Cell Signaling and Tissue Morphogenesis. In: *Current Topics in Developmental Biology* (ed Yap AS). Academic Press (2015).
4. Li KN, Chovatiya G, Ko DY, Sureshbabu S, Tumber T. Blood endothelial ALK1-BMP4 signaling axis regulates adult hair follicle stem cell activation. *EMBO J*, e112196 (2023).
5. Li KN, Jain P, He CH, Eun FC, Kang S, Tumber T. Skin vasculature and hair follicle cross-talking associated with stem cell activation and tissue homeostasis. *Elife* **8**, (2019).
6. Donovan P, *et al.* Endovascular progenitors infiltrate melanomas and differentiate towards a variety of vascular beds promoting tumor metastasis. *Nat Commun* **10**, 18 (2019).
7. Lukowski SW, *et al.* Single-Cell Transcriptional Profiling of Aortic Endothelium Identifies a Hierarchy from Endovascular Progenitors to Differentiated Cells. *Cell Rep* **27**, 2748-2758 e2743 (2019).
8. Patel J, *et al.* Functional Definition of Progenitors Versus Mature Endothelial Cells Reveals Key SoxF-Dependent Differentiation Process. *Circulation* **135**, 786-805 (2017).
9. Duong CN, Vestweber D. Mechanisms Ensuring Endothelial Junction Integrity Beyond VE-Cadherin. *Front Physiol* **11**, 519 (2020).
10. Gavard J. Endothelial permeability and VE-cadherin: a wacky comradeship. *Cell Adh Migr* **7**, 455-461 (2013).
11. Orsenigo F, *et al.* Phosphorylation of VE-cadherin is modulated by haemodynamic forces and contributes to the regulation of vascular permeability in vivo. *Nat Commun* **3**, 1208 (2012).
12. Liu Q, Wang X, Yi S. Pathophysiological Changes of Physical Barriers of Peripheral Nerves After Injury. *Front Neurosci* **12**, 597 (2018).
13. Liu JA, Yu J, Cheung CW. Immune Actions on the Peripheral Nervous System in Pain. *Int J Mol Sci* **22**, (2021).

14. Katagata Y, Takeda H, Ishizawa T, Hozumi Y, Kondo S. Occurrence and comparison of the expressed keratins in cultured human fibroblasts, endothelial cells and their sarcomas. *J Dermatol Sci* **30**, 1-9 (2002).
15. Miettinen M, Fetsch JF. Distribution of keratins in normal endothelial cells and a spectrum of vascular tumors: implications in tumor diagnosis. *Hum Pathol* **31**, 1062-1067 (2000).
16. Traweek ST, Liu J, Battifora H. Keratin gene expression in non-epithelial tissues. Detection with polymerase chain reaction. *Am J Pathol* **142**, 1111-1118 (1993).

REVIEWERS' COMMENTS

Reviewer #1 (Remarks to the Author):

The authors have addressed my major concerns by deleting Alk gene in Cdh5+/Krt19+ cell population using K19-CreER driver. They found that laminin staining intensity in tdTomato reporter-positive cells was decreased in the mutants. Given that the basement membrane seems to serve as a major microenvironment for both Cdh5+/Krt19+ cells and peripheral nerves as demonstrated by the authors' immunostaining analysis in Figure 5, this reduction could have profound impacts on the function of these cell populations. Indeed, increased branching structures were observed in tdTomato+ nerves/Schwann complex, suggesting the involvement of the Cdh5+/Krt19+ cell population in the regulation of skin nerve structures.

Furthermore, the authors have rewritten the title and abstract, aligning them more closely with their new findings and ensuring greater specificity and coherency.

As a result, the reviewer believes that the quality of this manuscript has significantly improved.

Reviewer #2 (Remarks to the Author):

In this revision, authors demonstrated good effort addressing my major concern and new Krt19-CreERT2;Alk1fl/fl mouse provides valuable addition to the story. I am supportive of publishing this manuscript at Nat Comms.

Minor/moderate suggestion - please consider simplifying Title to make it more accessible to general readership. I suggest removing at least some specific terms and use general terms instead, that do not require prior knowledge. For instance:

Hair growth-responsive mesenchymal cells regulate cyclical remodeling of peripheral nerve branching in mouse skin.

Likewise, I think that current Abstract does not sufficiently highlight the key findings and at least parts of it can be rewritten and improved.

Reviewer #3 (Remarks to the Author):

In the revision, authors have addressed some of my concerns and questions. However, the key question of specific evaluation of non-endothelial VE-cad expressing perineurial cells in skin is not performed in revision. It would be important to understand its specific function of Cdh5+/Krt19 cells during adult nerve homeostasis.

Response to Reviewers' Comments

Reviewer #1 (Remarks to the Author):

The authors have addressed my major concerns by deleting Alk gene in Cdh5+/Krt19+ cell population using K19-CreER driver. They found that laminin staining intensity in tdTomato reporter-positive cells was decreased in the mutants. Given that the basement membrane seems to serve as a major microenvironment for both Cdh5+/Krt19+ cells and peripheral nerves as demonstrated by the authors' immunostaining analysis in Figure 5, this reduction could have profound impacts on the function of these cell populations. Indeed, increased branching structures were observed in tdTomato+ nerves/Schwann complex, suggesting the involvement of the Cdh5+/Krt19+ cell population in the regulation of skin nerve structures.

Furthermore, the authors have rewritten the title and abstract, aligning them more closely with their new findings and ensuring greater specificity and coherency.

As a result, the reviewer believes that the quality of this manuscript has significantly improved.

Thank you. We are pleased that you found improvement in the manuscript's quality.

Reviewer #2 (Remarks to the Author):

In this revision, authors demonstrated good effort addressing my major concern and new Krt19-CreERT2;Alk1fl/fl mouse provides valuable addition to the story. I am supportive of publishing this manuscript at Nat Comms.

Minor/moderate suggestion - please consider simplifying Title to make it more accessible to general readership. I suggest removing at least some specific terms and use general terms instead, that do not require prior knowledge. For instance:

Hair growth-responsive mesenchymal cells regulate cyclical remodeling of peripheral nerve branching in mouse skin.

Likewise, I think that current Abstract does not sufficiently highlight the key findings and at least parts of it can be rewritten and improved.

We have changed the title and abstract one more time in an effort to make it more accessible and better summarize our broad findings.

Reviewer #3 (Remarks to the Author):

In the revision, authors have addressed some of my concerns and questions. However, the key question of specific evaluation of non-endothelial VE-cad expressing perineurial cells in skin is not performed in revision. It would be important to understand its specific function of Cdh5+/Krt19 cells during adult nerve homeostasis.

The editors informed us that: *“Regarding the comment of Reviewer #3, based on our editorial assessment after consulting with Reviewers #1 and #2, the key conclusions are sufficiently supported by the data.”*